# Grow and Merge: A Unified Framework for Continuous Categories Discovery

**Xinwei Zhang**[1],* **Jianwen Jiang**[2]*, **Yutong Feng**[2], **Zhi-fan Wu**[2], **Xibin Zhao**[1]†,
**Hai Wan**[1], **Mingqian Tang**[2], **Rong Jin**[2], **Yue Gao**[1,3]†
[1]BNRist, KLISS, School of Software, Tsinghua University
[2]Alibaba Group
[3]THUIBCS, BLBCI, Tsinghua University
`xinwei-z21@mails.tsinghua.edu.cn`
`{jianwen.jjw,fengyutong.fyt,wuzhifan.wzf,mingqian.tmq,jinrong.jr}`
`@alibaba-inc.com, {zxb,wanhai,gaoyue}@tsinghua.edu.cn`

## Abstract

Although a number of studies are devoted to novel category discovery, most of them assume a static setting where both labeled and unlabeled data are given at once for finding new categories. In this work, we focus on the application scenarios where unlabeled data are continuously fed into the category discovery system. We refer to it as the **Continuous Category Discovery** (**CCD**) problem, which is significantly more challenging than the static setting. A common challenge faced by novel category discovery is that different sets of features are needed for classification and category discovery: class discriminative features are preferred for classification, while rich and diverse features are more suitable for new category mining. This challenge becomes more severe for dynamic setting as the system is asked to deliver good performance for known classes over time, and at the same time continuously discover new classes from unlabeled data. To address this challenge, we develop a framework of **Grow and Merge** (**GM**) that works by alternating between a growing phase and a merging phase: in the growing phase, it increases the diversity of features through a continuous self-supervised learning for effective category mining, and in the merging phase, it merges the grown model with a static one to ensure satisfying performance for known classes. Our extensive studies verify that the proposed GM framework is significantly more effective than the state-of-the-art approaches for continuous category discovery.

## 1 Introduction

Human beings are good at grouping objects into category through clustering, and the definition of categories are continuously expanding and updated over time. Recent developments of intelligent visual systems can not only distinguish pre-defined categories [1–5], but also discover new categories from unlabeled data, a task that is known as *novel category discovery* [6–8].

Existing works on novel category discovery are limited to the *static* setting, where both labeled data (by known classes) and unlabeled data (with potential unknown categories) are given at once. In contrast, for real-world applications, unlabeled data are continuously fed into the system for discovering new categories, making it a significantly more challenging problem. Besides, current studies for novel category discovery often assume that all the unlabeled data belong to the unknown

---

*Equal contributions.
†Corresponding authors.

36th Conference on Neural Information Processing Systems (NeurIPS 2022).

new categories, which is generally not true in real applications. In this work, we examine the dynamic setup of novel category discovery where the system was initially given a set of data labeled by known classes, and unlabeled data are continuously streamed into the system for discovering new classes. The system is requested to consistently yield satisfying performance for known classes, and at the same time, dynamically discover new categories from the streaming unlabeled data. We refer to it as **Continuous Category Discovery**, or **CCD** for short.

We illustrate the process of CCD in Figure 1. It is comprised of two main stages: the *initial stage* where a classification model is trained by a set of labeled examples, and the *continuous category discovery stage* where new categories are continuously discovered from a stream of unlabeled data belonging to both known and unknown classes. A intuitive approach to address the dynamic nature of CCD is to combine the existing methods for open-set recognition [9–11], novel category discovery [12, 6, 13], and incremental learning [14, 15]. This is however insufficient because our learning system has to accomplish two tasks at the same time, *i.e.*, accurately classify instances into the known classes, and discover new categories from an unlabeled data stream. It turns out that these two task models usually produce different types of features: discriminative features on known classes are preferred by classification model, while rich and diverse features are critical for identifying new classes, as illustrated in Figure 2. A simple combination of novel category discovery and incremental learning will fail to address the trade-off consistently over time, which is further verified by our empirical studies.

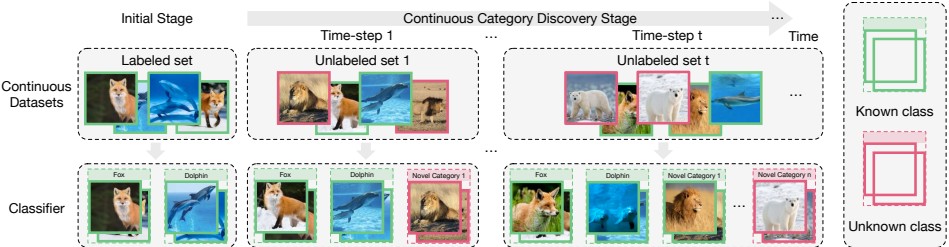

Figure 1: **Overview of the Continuous Category Discovery (CCD).** The continuous data stream is mixed with unlabeled samples from both known and novel categories. CCD requires to distinguish known categories, discover novel categories and merge the discovered categories into known set.

To address the challenge of continuous category discovery, we propose a framework of **Grow** and **Merge**, or **GM** for short. After pre-training a static model $\mathcal{A}$ over the labeled data, we will update model $\mathcal{A}$ with respect to unlabeled data stream by alternating between the growing phase and the merging phase: in the growing phase, we will increase the diversity of features by continuously training our model over received unlabeled data through a combination of supervised and self-supervised learning; in the merging phase, we will merge the grown model with the static one by taking a weighted combination of both models. By alternating between the growing and merging phases, we are able to maintain a good performance for known classes, and at same time, the power of discovering new categories. This is clearly visualized in Figure 2, where the first two panels show that existing approaches can do well on one of the two tasks but not both, and last panel shows that features learned by the proposed GM framework works well for both tasks.

Finally, one of the common issues with continuous training is catastrophic forgetting. To alleviating the forgetting effect as we are growing the number of categories over time, we maintain a small set of labeled samples from known categories and pseudo-labeled samples from novel categories. These selected examples are used in the growing phase to expand feature diversity for effective category discovery. Extensive experimental results show that our proposed method consistently shows satisfying performance under multiple practical scenarios compared with existing methods.

The main contributions of this paper are summarized as follows:

- We study a new problem named continuous category discovery, or CCD, which better reflects the challenge of category discovery in the wild. It needs to simultaneously maintain a good performance for known categories and the ability of discovering novel categories.

- We propose a framework of grow and merge, or GM, for CCD, that is able to resolve the conflicts between the classification task and the task of discovering new categories.

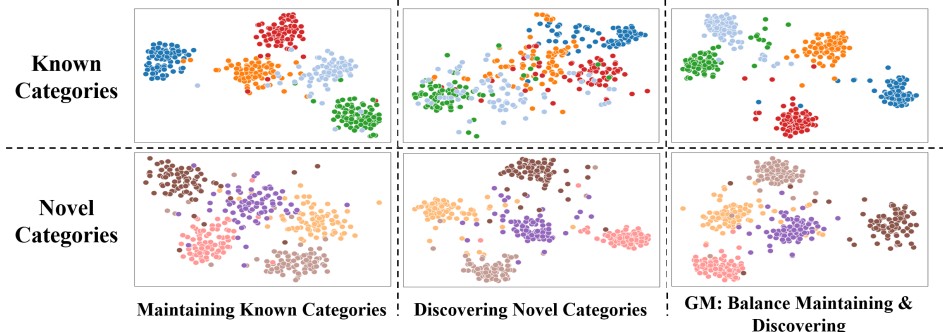

Figure 2: Features visualization of model A trained on known categories, model B trained for novel category discovery (based on model A) and the proposed GM model.

- We conduct experiments under four different settings to fully investigate the scenarios of novel category discovery in the wild. The proposed method shows less forgetting of known categories and better performance for category discovery compared to existing methods.

## 2 Problem Definition

In this section, we formulate the problem of Continuous Category Discovery (CCD). There are two main stages of the CCD problem, *i.e.*, **Initial Stage** and **Continuous Category Discovery Stage**. The settings of each stage are introduced, together with the evaluation metrics of CCD problem.

### 2.1 Setting of Continuous Category Discovery

During the initial stage, a labeled training dataset $\mathcal{D}^0_{train} = \{(\boldsymbol{x}^0_i, y^0_i)\}^{N^0}_{i=1}$ is provided to train the model on the initial known category set $\mathcal{C}^0 = \{1, 2, ..., K^0\}$, where $\boldsymbol{x}^0_i$ is the initial training data, and $y^0_i \in \mathcal{C}^0$ is the corresponding label. The model is expected to classify categories in $\mathcal{C}^0$, and learn meaningful representation from the labels' semantic information.

During the continuous category discovery stage, a serial of unlabeled training datasets $\{\mathcal{D}^t_{train}\}^T_{t=1}$ are sequentially provided, where $\mathcal{D}^t_{train} = \{\boldsymbol{x}^t_i\}^{N^t}_{i=1}$ indicates the dataset at time $t$. Though unlabeled, we denote the known or potential appeared categories until time $t$ as $\mathcal{C}^t = \{1, 2, ..., K^t\}$. The model is expected to discover newly appeared unknown categories from $\mathcal{D}^t_{train}$, store the representations of the discovered novel categories and maintain the knowledge of the known categories.

### 2.2 Evaluation Metrics

At each time-step $t$, we evaluate the classification performance on the test dataset $\mathcal{D}^t_{test} = \{(\boldsymbol{x}^s_i, y^s_i)|s \leq t\}$, containing test samples from all the known or previously discovered categories. For the newly appeared unknown categories, we evaluate the novel category discovery performance. The **maximum forgetting metric** $\mathcal{M}_f$ and the **final discovery metric** $\mathcal{M}_d$ are designed for evaluation. To evaluate the performance of the clustering assignments, we follow the standard practise [6, 16] to adopt clustering accuracy on the known categories and the novel categories, denoted as $\text{ACC}^t_{known}$ and $\text{ACC}^t_{novel}$, respectively. The maximum forgetting $\mathcal{M}_f$ is defined as the maximum value of the differences between $\text{ACC}^0_{known}$ and $\text{ACC}^t_{known}$ for every $t$ and the final discovery $\mathcal{M}_d$ is defined as the final cluster accuracy on novel categories, *i.e.*, $\mathcal{M}_d = \text{ACC}^T_{novel}$. See more details in the supplementary materials.

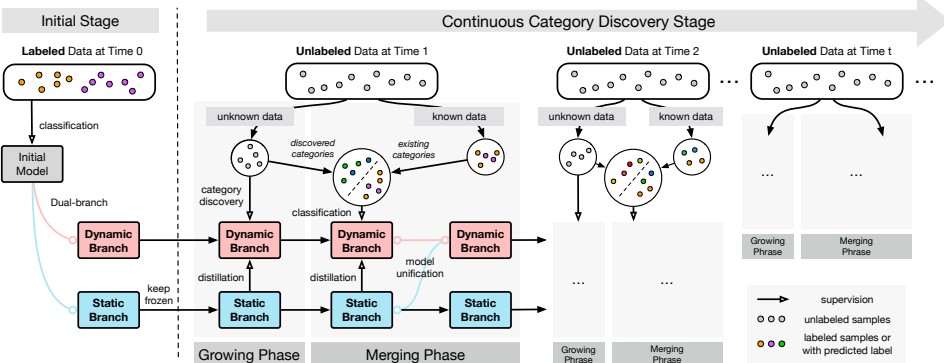

Figure 3: **The proposed Grow and Merge (GM) Framework.** There exist growing and merging phases for each time-step, where a dynamic branch is freely updated for discovering new classes in the growing phase, and then unified with a frozen static branch in the merging phase for maintaining classification on all appeared classes.

## 3 Method

### 3.1 An Overview of Grow and Merge Framework

The CCD problem requires both maintaining the performance on known categories and discovering novel categories under the continuous learning scheme. For the compatibility of the two tasks, we propose a **Grow and Merge (GM)** Framework, which is illustrated in Figure 3.

During the initial stage, the model $\phi^0(\cdot)$ is firstly pre-trained on the initial dataset with labeled data in both supervised and self-supervised manner. An *exemplars* set $P_k = \{\mathbf{p}_{k,i}\}$ is constructed as well as the *prototypes*. The exemplar set contains representative samples for each class while the prototype $\mu_k$ is the averaged feature vector of the exemplars from the $k$-th class. The exemplar set is constructed by iteratively picking several typical samples with the principle of maintaining the closest center to the prototype $\mu_k$, *i.e.* $\mathbf{p}_{k,i} \leftarrow \arg\min_{\mathbf{x}} \| \mu_k - \frac{1}{i}[\phi^0(\mathbf{x}) + \Sigma_{j=1}^{i-1}\phi^0(\mathbf{p_{k,j}})] \|$. The definitions of prototype and exemplars are adapted from iCaRL [15], which enables us to use a non-parametric classifier that predicts label of a query sample from its nearest prototype's label.

During the continuous category discovery stage, a dual-branch architecture with a static branch $\phi_S(\cdot)$ and a dynamic branch $\phi_D(\cdot)$ is introduced for training, where both branches are initialized by the pre-trained model $\phi^0(\cdot)$. At each time-step, GM includes two phases: **Growing** and **Merging**. In the growing phase, the novel category data are filtered out from the continuous data and the dynamic branch is trained for novel category discovery. Then the merging phase fuses the newly discovered and known categories into a single model for unified classification. Knowledge from the frozen static branch is distilled into the dynamic one to avoid forgetting previous classes.

It is noted that though GM maintains two branches during training, only one branch is needed during testing.

Experimental results show significant performance of the GM framework, where the dual-branch architecture and knowledge distillation succeed to balance the feature conflicts between continuous classification and novel categories discovery, and show better performance on both sub-tasks.

### 3.2 Growing: Enrich Features with Unlabeled Data

To achieve novel category discovery from the data stream with both labeled and unlabeled samples from known and unknown classes, we firstly filter out data of unknown classes with **novelty detection**, and then conduct **novel category discovery** based on both the new data and the previous model.

**Novelty Detection** aims to identify the samples belonging to the novel categories in the current training dataset $\mathcal{D}_{train}^t$, since there may also exist samples from previously known categories. With the maintained prototypes of known categories, the feature vectors of samples from known categories would lay closer to at least one prototype with a high probability. Then a novel distance

$d_{\text{novel}}$ is defined as the distance between the sample $\mathbf{x}_i^t$ and its nearest prototype, *i.e.* $d_{\text{novel}}(\mathbf{x}_i^t) = \min_{1 \le k \le K^t} d(\mu_k, \phi_D^t(\mathbf{x}_i^t))$, where $d(\cdot, \cdot)$ is the distance function. Samples with $d_{\text{novel}}$ larger than a threshold $\epsilon$ are filtered out as novel samples for the following process.

**Novel Categories Discovery and Learning from Static Branch**. We optimize the dynamic branch $\phi_D^t(\cdot)$ for novel category discovery, guided by both the novel samples and previously learned knowledge for rich feature representation. (i) For learning and discovering from the novel samples, we adopt an unsupervised representation learning method [6]. A cluster head $g(\cdot)$ is learned to assign samples $\hat{\mathbf{x}}_i^t$ to the clusters with novel category indices $\{K^{t-1} + 1, ..., K^t\}$. Such an assignment is supervised by the similarity $s_{ij}$ between samples $\hat{\mathbf{x}}_i^t$ and $\hat{\mathbf{x}}_j^t$ with a binary cross entropy loss, *i.e.*

$$\mathcal{L}_{\text{BCE}} = -\frac{1}{\hat{N}^t} \sum_{i=1}^{\hat{N}^t} \sum_{j=1}^{\hat{N}^t} \left( s_{ij} \log \mathbf{c}_i^{t\top} \mathbf{c}_j^t + (1 - s_{ij} \log(1 - \mathbf{c}_i^{t\top} \mathbf{c}_j^t)) \right), \tag{1}$$

where $\hat{N}^t$ is the number of novel category samples at time $t$ (specified or estimated). The similarity $s_{ij}$ is measured by the Winner-Take-All hash (WTA), which equals one when samples share the same top-$k$ channels of their feature vector, otherwise equals zero.

(ii) For learning from the static branch $\phi_D^t(\cdot)$ with previously learned knowledge, we introduce a *static-dynamic distillation loss* to transfer the representation learned from labeled data in the static branch into the dynamic one:

$$\mathcal{L}_{\text{SD}} = \frac{1}{N^t} \sum_i^{N_t} d\left(\mathbf{z}_i^t, \mathbf{z}_i'^t\right) = \frac{1}{N^t} \sum_i^{N_t} d\left(\phi_S(\mathbf{x}_i^t), \phi_D^t(\mathbf{x}_i^t)\right) \tag{2}$$

The joint loss $\mathcal{L}_{\text{BCE}}$ and $\mathcal{L}_{\text{SD}}$ enables the dynamic branch to discovering novel categories with richer feature representation, and the cluster head $g(\cdot)$ could assign new labels to the novel category data $\hat{\mathbf{x}}_i^t$.

## 3.3 Merging: Unification of Categories and Branches

After the Growing phase, both categories and models contain two parts for the known and novel categories, respectively. The Merging phase reunifies them together for all the appeared categories.

### 3.3.1 Category Unification: Continuous Learning for Newly Discovered Categories

To merge the novel categories discovered in Growing phase into the classifier, we firstly **sift samples** to construct the exemplar sets of the novel categories, then refine the model representation using a **pseudo label representation learning** for classification on both known and novel classes.

**Sample Sifting** aims to filter out incorrectly-assigned samples based on their cluster confidence. The assigned class indices by $g(\cdot)$ are denoted as the pseudo-labels of the novel samples, which may inevitably have errors and lead to less reliable prototypes of the novel categories. We sift out these potentially noisy samples via the local sample density, with the assumption that samples from the same class would lay tightly. Suppose $\mathcal{N}_j(\hat{\mathbf{z}}_i^t)$ indicates the $j$ nearest neighbors of $\hat{\mathbf{z}}_i^t$ from the same class. The local sample density $G_j(\hat{\mathbf{z}}_i^t)$ is defined as the distance between $\hat{\mathbf{z}}_i^t$ and its $j$-th nearest neighbors, *i.e.* $G_j(\hat{\mathbf{z}}_i^t) = \max_{\mathbf{z} \in \mathcal{N}_j(\hat{\mathbf{z}}_i^t)} d(\hat{\mathbf{z}}_i^t, \mathbf{z})$. Samples with larger $G_j$ are sifted out, then the left samples are denoted as $\tilde{\mathbf{x}}_i^t$ with pseudo labels $\tilde{c}_i^t$.

**Pseudo Label Representation Learning.** Based on the sifted novel samples of the $k$-th novel category, we firstly construct the exemplar set $P_k = \{\mathbf{p}_{k,i}\}$. Without knowing the prototype $\mu_k$, *i.e.* the explicit center of the $k$-th category, we estimate it with the averaged vectors of sifted samples. For the purpose of continuous classification on all appeared classes, it is required to tighten the sparse representation distribution learned in the growing phase. Pseudo Label Learning (PLL) loss is introduced here. Compared with commonly-used Cross Entropy loss, PLL does not require an explicit classifier and thus does not modify the deployed model structure during the continuous phase. The loss function $\mathcal{L}_{\text{PLL}}$ aims to pull exemplars pull the exemplars $\mathbf{p}_k$ and the corresponding prototypes $\mu_k$ closer, and push $\mathbf{p}_k$ away from the other prototypes $\mu_j (j \neq k)$:

$$\mathcal{L}_{\text{PLL}} = -\frac{1}{K^t} \sum_{k=1}^{K^t} \frac{1}{|P_k|} \sum_{i=1}^{|P_k|} \log \frac{\exp(\phi_D^t(\mathbf{p}_{k,i}) \cdot \mu_k / \tau)}{\sum_{j=1}^{K^t} \exp(\phi_D^t(\mathbf{p}_{k,i}) \cdot \mu_j / \tau)}, \tag{3}$$

where $\tau$ is a temperature hyper-parameter. To avoid catastrophic forgetting, based on data in the latest exemplars, the static-dynamic distillation loss in Eq. 2 is also employed here.

### 3.3.2 Branch Unification: Merging Multiple Branches into One Unified Model

During the above process, the dynamic branch is optimized to classify all appeared categories. In order to reduce computation cost and further utilize the representation of the static branch trained from labeled data, we merge the two branches into a single one. In experiments, we found such a branch merging can further improve the recognition performance. To fuse the representation of $\phi_S$ and $\phi_D^t$, the exponential moving average strategy (EMA) is applied:

$$\theta_{\phi_D^t} \longleftarrow \alpha\theta_{\phi_S} + (1-\alpha)\theta_{\phi_D^t}, \tag{4}$$

where $\alpha$ is the momentum hyper-parameter to control the weight of fusion, generally valued close to 1 ($\alpha = 0.99$ in this paper).

### 3.4 Estimating the Number of Novel Categories

Grow and Merge framework is proposed to solve the CCD problem. GM is a generic framework which focuses on how to implementing continuous classfication and novel classes discovery at the same time. Estimating the number of novel categories has been studied in literatures including [12, 16], which can be easily employed in GMNet. For each time step $t$, the representations of the coming data $\phi(\mathbf{x}^t)$ is mixed together with the exemplars $P$. The semi-Kmeans algorithm is applied on these mixed data with different number of clusters, and the cluster accuracy on the exemplars are evaluated based on the pseudo labels of the exemplars. The number of the clusters with the highest cluster accuracy is estimated as the number of novel categories.

## 4 Experiments

In this section, we design experiments in different scenarios to simulate the varying cases of real-world applications, which enables us to fully investigate the performance and flexibility of the proposed method. Two main evaluation metrics, namely, maximum forgetting $\mathcal{M}_f$ and final discovery $\mathcal{M}_d$ are employed to measure the performance. CIFAR-100 [17], CUB-200 [18] and ImageNet-100 [19, 20] representing the small, middle and large scale datasets are used in the experiments.

**Experimental Scenarios**.We formulate four different scenarios for experiments as follows. (1) *Class Incremental Scenario (CI)*: the data are only drawn from novel categories. (2) *Data Incremental Scenario (DI)*: the data are only drawn from the known categories. (3) *Mixed Incremental Scenario (MI)*: the data are drawn from both novel and known categories. (4) *Semi-supervised Mixed Incremental Scenario (SMI)*: the data are drawn from both novel and known categories, and a portion of the data are labeled, which is closed to the real-world application.

Table 1: The content of $\mathcal{D}_{train}^t$ under different scenarios DI, CI, MI and SMI.

|  | Known Categories | Novel Categories | Labeled data | Unlabeled data |
|---|---|---|---|---|
| CI | ✗ | ✓ | ✗ | ✓ |
| DI | ✓ | ✗ | ✗ | ✓ |
| MI | ✓ | ✓ | ✗ | ✓ |
| SMI | ✓ | ✓ | ✓ | ✓ |

The settings of these scenarios are summarized in Table 1. We mainly focus on the standard scenario CI in this paper, while the experimental results on DI, MI, and SMI are reported to demonstrate the flexibility of the proposed method. Each scenario contains $T = 3$ time-steps of the continuous learning stage. Details of the scenarios could be found in supplementary materials.

**Implementation Details**. For all experiments, we train the model 100 epochs using $\mathcal{L}_{\text{BCE}} + \mathcal{L}_{\text{SD}}$ loss and 100 epochs using $\mathcal{L}_{\text{BCE}} + \mathcal{L}_{\text{SD}} + \mathcal{L}_{\text{PLL}}$ loss for each time-step $t$. The $\alpha$ for EMA is set to 0.99. The novelty detection threshold $\epsilon$ is set to 0.6. 15 neighbors are selected for each sample during sample sifting and the top 50% samples with larger local density are sifted out. The

Table 2: Experimental results under CI scenario (mean±std.).

| Method | CIFAR-100 | | CUB-200 | | ImageNet-100 | |
|---|---|---|---|---|---|---|
| | $\mathcal{M}_f \downarrow$ | $\mathcal{M}_d \uparrow$ | $\mathcal{M}_f \downarrow$ | $\mathcal{M}_d \uparrow$ | $\mathcal{M}_f \downarrow$ | $\mathcal{M}_d \uparrow$ |
| *lower-bound of $\mathcal{M}_f$ (offline K-Means)* | 5.37±0.70 | 16.95±1.04 | 2.94±1.03 | 14.29±0.47 | 3.03±5.47 | 18.20±0.58 |
| *upper-bound of $\mathcal{M}_d$ (offline AutoNovel)* | 5.12±0.84 | 42.27±4.97 | 7.57±0.77 | 16.13±0.37 | 6.81±0.98 | 34.18±1.77 |
| AutoNovel (online) | 66.76±1.56 | 29.65±4.60 | 60.66±0.20 | 18.08±0.84 | 63.00±0.42 | 5.83±1.81 |
| DRNCD (online) | 64.99±1.89 | 8.53±2.35 | 42.74±0.28 | 6.30±0.93 | 47.02±2.18 | 5.00±1.37 |
| AutoNovel (online) + LwF | 23.94±1.54 | 27.84±0.57 | 59.82±0.58 | 23.23±4.81 | 61.31±0.37 | 7.59±0.55 |
| DRNCD (online) + LwF | 26.87±0.15 | 3.57±0.61 | 45.25±0.37 | 5.78±0.51 | 30.88±1.57 | 6.78±1.16 |
| iCaRL (fixed exemplars) + LwF | 34.79±0.29 | - | 42.02±0.31 | - | 10.04±0.28 | - |
| **GM (Ours)** | **9.87±0.25** | **35.97±1.28** | **19.46±1.63** | **24.97±1.81** | **8.30±0.42** | **27.24±0.83** |

| Method | Stanford-Cars | | FGVC-Aircraft | | ImageNet-200 | |
|---|---|---|---|---|---|---|
| | $\mathcal{M}_f \downarrow$ | $\mathcal{M}_d \uparrow$ | $\mathcal{M}_f \downarrow$ | $\mathcal{M}_d \uparrow$ | $\mathcal{M}_f \downarrow$ | $\mathcal{M}_d \uparrow$ |
| *lower-bound of $\mathcal{M}_f$ (offline K-Means)* | 2.41±0.59 | 10.23±0.32 | 3.86±0.71 | 15.10±1.25 | 3.16±0.76 | 15.28±0.67 |
| *upper-bound of $\mathcal{M}_d$ (offline AutoNovel)* | 11.24±0.59 | 20.69±3.28 | 6.95±2.40 | 27.67±0.86 | 5.98±0.67 | 24.62±0.47 |
| AutoNovel (online) | 72.07±0.17 | 16.71±1.40 | 53.23±0.60 | 26.93±1.48 | 60.79±0.61 | 10.21±2.52 |
| DRNCD (online) | 50.70±3.29 | 2.39±0.84 | 38.89±4.15 | 5.35±0.07 | 43.91±1.37 | 4.39±1.20 |
| AutoNovel (online) + LwF | 71.52±0.33 | 20.31±0.69 | 44.63±1.37 | 8.68±5.67 | 61.03±0.37 | 9.77±1.00 |
| DRNCD (online) + LwF | 27.40±0.50 | 3.88±0.63 | 38.89±4.15 | 5.61±3.00 | 27.40±0.84 | 6.71±1.55 |
| iCaRL (fixed exemplars) + LwF | 24.31±0.64 | - | 22.60±1.19 | - | 10.55±0.52 | - |
| **GM (Ours)** | **17.66±0.70** | **25.77±0.54** | **8.21±0.27** | **29.65±1.63** | **7.50±1.60** | **19.33±0.46** |

temperature $\tau$ used in $\mathcal{L}_{\text{PLL}}$ is set to 0.1. Mean Squared Error (MSE) Consistency Loss $\mathcal{L}_{\text{MSE}} = \frac{1}{N^t} \sum_i^{N^t} ||\phi_D^t(\boldsymbol{x}_i^t) - \phi_D^t(\boldsymbol{x}_i'^t)||_2^2$ is also used during the continuous learning following AutoNovel [6], where $\boldsymbol{x}_i'^t$ denotes the augmentation of $\boldsymbol{x}_i^t$. More details could be found in supplementary materials.

## 4.1 Comparisons with SOTA methods on CI scenario

We first investigate the performance of the proposed method under CI scenario. CIFAR-100, CUB-200, ImageNet-100, Stanford-Cars, FGVC-Aircraft, and ImageNet-200 are used in this experiment. The maximum forgetting $\mathcal{M}_f$ and final discovery $\mathcal{M}_d$ are reported in Table 2. Since there are few works about continuous novel category discovery and recent works about incremental learning may not fit the unsupervised scenario, we compare the proposed method with the lower bound of $\mathcal{M}_f$ and the upper bound of $\mathcal{M}_d$, two recent methods for novel category discovery, *i.e.*, AutoNovel [6] and DRNCD [16], and the combinations of the novel category discovery methods and continuous learning methods.

It is noticed that there is not an exact lower/upper bound method under the CI scenario. Therefore, we conduct two methods under the **offline setting**, where all of the data containing both known and novel categories are available in a single stage. We present the results of K-means [21] as a reference to the lower bound of $\mathcal{M}_f$, and the original AutoNovel method as the upper bound of $\mathcal{M}_d$. It should be enhanced that the lower/upper bound results are for reference and could not exactly represents the best performance on both $\mathcal{M}_f$ and $\mathcal{M}_d$. Since recent novel category discovery and incremental learning methods usually contain multiple stages, we chose AutoNovel and LwF [14] and combine them for comparison based on the flexibility of implementation. It is observed that AutoNovel with LwF greatly improves the $\mathcal{M}_f$ and $\mathcal{M}_f$ compared to the online version of AutoNovel. However, such a simple combination not completely solves the feature conflict caused by the two different tasks. The proposed GM framework divides and conquers the conflict of features and achieves the best results among the compared methods (that conduct classification and discovery training in one single phase) on all datasets, indicating the effectiveness of the two phases of growing and merging.

Table 3: Experimental results under DI scenario. (mean±std.)

| Method | $\mathcal{M}_f \downarrow$ |
|---|---|
| Original Model | 0 |
| Pseudo iCaRL | 9.51±0.35 |
| DeepClustering [22] | 1.61±0.44 |
| AutoNovel* [23, 24] | 0.30±0.07 |
| GM | **-23.68±0.27** |

Table 4: Experimental results under MI and SMI scenario. (mean±std.)

| Method | MI | | SMI | |
|---|---|---|---|---|
| | $\mathcal{M}_f \downarrow$ | $\mathcal{M}_d \uparrow$ | $\mathcal{M}_f \downarrow$ | $\mathcal{M}_d \uparrow$ |
| AutoNovel [6] | 63.25±0.19 | 5.74±0.17 | 60.98±1.27 | 21.61±3.94 |
| DRNCD [16] | 43.81±1.71 | 5.12±0.98 | 56.27±1.13 | 47.67±1.66 |
| AutoNovel + LwF [6, 14] | 61.89±0.87 | 5.59±1.99 | 61.45±2.43 | 31.84±1.06 |
| DRNCD + LwF [16, 14] | 35.44±4.61 | 4.10±0.99 | 62.38±1.01 | **52.53±1.92** |
| GM w/o. novelty detection | **4.74±0.78** | 10.61±3.37 | **8.43±0.45** | 31.42±0.76 |
| GM | 9.65±0.32 | **30.58±1.13** | 9.41±0.31 | 35.52±1.21 |

Table 5: Estimation of the number of novel class in each time step, as well as the corresponding performances using the estimated number.

| | #Classes ($t=2$) | #Classes ($t=3$) | #Classes ($t=4$) | $\mathcal{M}_f$ | $\mathcal{M}_d$ |
|---|---|---|---|---|---|
| CIFAR-100 (CI) | 11 | 13 | 13 | 9.91±0.32 | 34.41±0.96 |
| ImageNet-100 (CI) | 12 | 13 | 13 | 9.19±0.51 | 26.26±1.27 |
| CIFAR-100 (MI) | 14 | 13 | 13 | 0.33±0.59 | 28.06±0.43 |
| CIFAR-100 (SMI) | 14 | 14 | 14 | 9.63±0.38 | 35.31±1.60 |

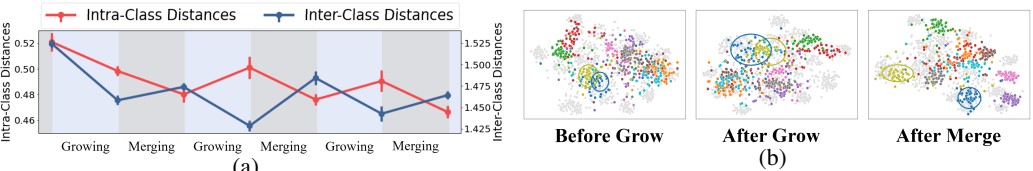

Figure 4: (a) The intra-/inter- class distance measured throughout the growing and merging phases. (b) Feature visualizations at different time of GM, where the colored dots denote previously appeared classes, and the gray dots denote the newly appeared classes without labels.

## 4.2 Extended Experiments on More Complicated Scenarios

In this part, we extend GM to more complicated scenarios: DI, MI, and SMI scenarios. Experiments are conducted on CIFAR-100 and the results are listed in Table 3 and Table 4. Due to there is no novel category in the incremental unlabeled data under DI, we remove the cluster head in the AutoNovel to form compared method AutoNovel$^*$. Incremental learning method iCaRL [15] combined with pseudo label assignment and unsupervised learning method DeepClustering combined with finetuning on incremental data are also chosen for comparisons. As shown in Table 3, after finetuning on the incremental unlabeled data, all comparison methods show the catastrophic forgetting effect, and the performances are even worse than the original model. GM avoids catastrophic forgetting and effectively utilizes continuous unlabeled data, further improving the final performance. In MI and SMI scenarios, incremental data are mixed with known and novel categories, which makes label assignment for incremental data more difficult. Nonetheless, as shown in Table 4, GM can still effectively avoid catastrophic forgetting and achieve superior performance (especially on $\mathcal{M}_f$) with the help of two phases of growing and merging, compared to the comparison methods.

## 4.3 Estimating the Number of Novel Categories

In this part, experiments with the unknown number of novel categories are conducted on CIFAR-100 and ImageNet-100 in CI, MI, and SMI scenario. The model should estimate the number of novel categories, discover the novel categories, and maintain the performance on the known categories during each time step. The experimental results are provided in Table 5. It could be found that GM without knowing the ground-truth number of novelty classes can still outperform other methods using the ground-truth number of novelty classes. GM model only has 1.33 and 2.47 decreases under CI scenario, and no more than 4.71 decreases under more challenging MI and SMI scenarios, which is acceptable.

## 4.4 Ablation Studies

Previous experimental results indicate Grow and Merge framework can better accomplish the continuous learning and the novel category discovery in multiple scenarios. Here we provide intra- and inter- class distance among features and the visualizations in different time of GM in Figure 4. It can be clearly observed that with the alternation of the growing and merging phases, the intra-class distance first increases and then decreases. The classes are relatively compact but overlapped with others before growing phase, loose after growing phase, and compact and separated with others after merging phase. The results are consistent to our motivation that the GM relaxes and tightens the features alternatively to resolve the feature conflicts caused by CCD task. The detailed analysis for all proposed key components of GM are provided. Ablation studies are conducted on the CIFAR-100

dataset under CI scenario, shown in Table 6 and Tabel 7. We also provide the experiments about estimating the number of novel categories in the supplementary material.

**Growing Phase**. We investigate the performance of introducing additional methods for representation learning in the growing phase. Combining with cross entropy loss (CE) of the exemplars degrades $\mathcal{M}_d$, which validates that relaxing the constraint of memorizing previous category information can facilitate category discovery. Thus, we further introduce more relaxed self-supervised methods. Combing SimCLR [3] also degrades $\mathcal{M}_d$, which could be attributed that pushing the representation of negative samples away hurts the performance. An alternative method SimSiam [25] achieves comparable performance without pushing negative samples, which also validates the importance of a relaxed representation in this phase.

**Merging Phase**. In the merge phase, PLL is employed to improve the representation. We evaluate the performance of the models replacing PLL with other representation learning methods. *CE* represents cross entropy loss with the pseudo label given by the prototypes and *SSL* denotes standard InfoNCE loss [3, 4]. The experimental results demonstrate that, though these alternative methods could perform competitively on known classes, they sacrifice the performance of discovering novel classes.

**Model Merging**. We evaluate the performance of the model without EMA. $\mathcal{M}_f$ increases for 55% and $\mathcal{M}_d$ decreases for 14%, which demonstrates the necessity of EMA. We also consider an alternative fuse method which assigns $\phi_S \leftarrow \phi_D^t$ after the update of $\phi_D^t$ by Eq. 4. $\mathcal{M}_d$ is significantly affected with a decrease of 33%. Therefore, we use the fixed $\phi_S$ in this paper.

**Static-Dynamic Branch**. The multi-branch aims to maintain known category information and discover novel categories on different branches. We first evaluate the performance when directly replacing the multi-branch with a single one. The results show that $\mathcal{M}_f$ increases dramatically, which demonstrates the importance of the multi-branch to preserve already learned categories. We further investigate the components of the static-dynamic branch. Removing the initialization of branch $\phi_D^0$ also degrades both $\mathcal{M}_f$ and $\mathcal{M}_d$. For the proposed loss $\mathcal{L}_{SD}$ aligning the feature space of $\phi_S$ and $\phi_D^t$, results show that the model without $\mathcal{L}_{SD}$ performs worse on both $\mathcal{M}_f$ and $\mathcal{M}_d$ for the separation of the feature spaces leading to the failure of the prototype mechanism. The model replacing $\mathcal{L}_{SD}$ with DINO [26] increases $\mathcal{M}_f$ for 3% and decreases $\mathcal{M}_d$ for 6.3%.

**Sifting in Merging**. Sifting aims to sift out samples with low cluster confidence given by the cluster head. Except for the criterion based on the adopted local geometry property, there are several potential criteria to sift samples. *Confidence Ranking (CR)* selects a portion of all samples with the highest score . *Conditional Confidence Ranking (CCR)* selects a portion of samples from each class with the highest score. *Confidence Filtering (CF)* selects the samples with the score higher than a given threshold. The results show that: (i) For $\mathcal{M}_f$, the model without sifting and CR performs worse, and CCR, CF and the local sample density based method performs at almost same level. (ii) For $\mathcal{M}_d$, the local sample density based method outperforms all of these compared methods.

**Hyperparameter Analysis of Static-Dynamic Branch**. We also evaluate the effect of $\alpha$ in Eq. 4, shown in Figure 5. The hyper-parameter $\alpha$ controls the importance of $\phi_S$ relative to $\phi_D^t$. For the model with only EMA, both $\mathcal{M}_f$ and $\mathcal{M}_d$ decrease with the increase of $\alpha$, since the model with higher $\alpha$ maintains more information from the known categories. The model with only $\mathcal{L}_{SD}$ (*i.e.*, the model w/o. EMA shown in Tabel 6) performs worse than the proposed full model, which demonstrates that only the constrain of the distance between the feature space of $\phi_S$ and $\phi_D^t$ is not enough to transfer the knowledge from $\phi_S$ to $\phi_D^t$. The performances of the models with both EMA and $\mathcal{L}_{SD}$ varies smoother, compared with the models with only EMA. Therefore, the models could achieve higher performances and be more robust by combining EMA and $\mathcal{L}_{SD}$.

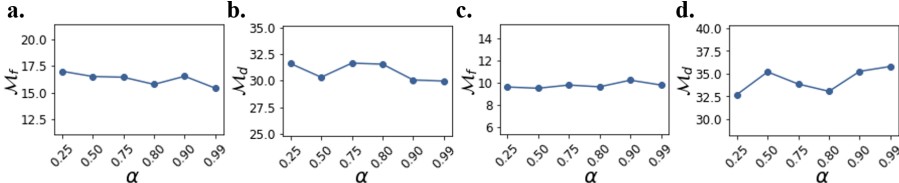

Figure 5: The variation comparison of the performance of the models by varying $\alpha$. **a**: $\mathcal{M}_f$ of the model with only EMA; **b**: $\mathcal{M}_d$ of the model with only EMA; **c**: $\mathcal{M}_f$ of the model with both EMA and $\mathcal{L}_{SD}$; **d**: $\mathcal{M}_d$ of the model with both EMA and $\mathcal{L}_{SD}$.

Table 6: Results of ablation studies on GM phases. (mean±std.)

| | | $\mathcal{M}_f \downarrow$ | $\mathcal{M}_d \uparrow$ |
|---|---|---|---|
| GM | | 9.87±0.25 | 35.97±1.28 |
| Growing Phase | WTA+$\mathcal{L}_{SD}$+CE | 9.54±0.04 | 31.37±0.19 |
| | WTA+$\mathcal{L}_{SD}$+SimCLR | 9.74±0.08 | 32.29±1.42 |
| | WTA+$\mathcal{L}_{SD}$+SimSiam | 9.56±0.06 | 34.81±1.06 |
| Merging Phase | CE + PLL | 8.83±0.30 | 34.48±1.09 |
| | CE | 7.27±0.25 | 31.95±0.81 |
| | SSL + PLL | 9.19±0.17 | 34.94±2.01 |
| | SSL | 7.55±0.28 | 31.03±1.82 |
| Model Merging | w/o. EMA | 17.72±2.12 | 31.18±2.15 |
| | Update $\phi_S$ | 11.75±1.35 | 24.34±2.66 |

Table 7: Results of ablation studies on Static-Dynamic Branch and Sifting in the Merging. (mean±std.)

| | | $\mathcal{M}_f \downarrow$ | $\mathcal{M}_d \uparrow$ |
|---|---|---|---|
| GM | | 9.87±0.25 | 35.97±1.28 |
| Static-Dynamic Branch | w/o. Multi-Branch | 33.46±0.65 | 33.80±1.17 |
| | w/o. Initialize $\phi_D^0$ | 26.91±1.16 | 31.81±2.27 |
| | w/o. $\mathcal{L}_{SD}$ | 15.78±0.42 | 31.71±1.88 |
| | DINO | 9.58±0.30 | 32.89±1.24 |
| Sifting in the Merging | w/o. Sifting | 10.00±0.15 | 31.68±0.45 |
| | Sifting (CR) | 9.91±0.19 | 33.18±1.46 |
| | Sifting (CCR) | 9.74±0.58 | 32.86±0.95 |
| | Sifting (CF) | 9.78±0.06 | 31.93±0.66 |

## 5   Related Work

**Novel Class Discovery** [6–8, 27] aims to discover unknown classes from unlabeled data by leveraging pairwise similarity [6, 28] or employing unsupervised clustering [13, 29]. Most of these methods assume that there is a pure unlabeled set that only contains samples from unknown categories, which means the class space of unlabeled data is completely disjoint with the labeled set. And at test time, samples are all from novel classes. These assumptions limit their applications in the real world. CCD aims to tackle novel class discovery under a more realistic situation in which the unlabeled data (i) are continuously confronted and (ii) contain both known and unknown classes.

**Continual Learning** needs the model to extend its ability to address new tasks when confronted with new data [30–38]. The main challenge of continual learning is to avoid catastrophic forgetting [39, 40], *i.e.*, the performance degradation on previously learned tasks when new ones are ingested. Existing methods can be categorized into replay methods [15, 41, 42], regularization-based methods [14, 43, 44] and parameter isolation methods [45–47]. Most of the methods assume there are available labeled data in each time-step. Nevertheless, the data in continuous streams are more likely to be unlabeled in reality, which means the category information is unknown and needs to be discovered.

Different from existing novel category discovery and continual learning, CCD requires the model to discover novel categories from the continuous unlabeled data while maintaining the classification performance on known and discovered classes.

## 6   Conclusion

In this paper, we study the continuous category discovery (CCD) problem where unlabeled data are continuously fed into the category discovery system. The main challenge of the CCD problem is that different sets of features are needed to classify known categories and discover novel categories. To address this challenge, we develop a framework called Grow and Merge that alternatively updates the model in the growing phase and the merging phase. Extensive experimental results demonstrate that GM outperforms the existing methods with the less forgetting effect of known categories and higher accuracy for category discovery.

## Acknowledgement

This research is sponsored in part by the Key-Area Research and Development Program of Guangdong Province under Grant 2020B010164001 and the NSFC Program (No. 62076146,62021002,U1801263,U20A6003,U19A2062,U1911401,62127803). This work was supported by Alibaba Group through Alibaba Innovative Research Program.

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
