# OpenReview forum: "Grow and Merge: A Unified Framework for Continuous Categories Discovery"
_NeurIPS.cc/2022/Conference — NeurIPS 2022 Accept_

### Official Review · Reviewer_WTKt · 2022-07-09

**Rating:** 5
**Confidence:** 5
**Soundness:** 3 good
**Presentation:** 3 good
**Contribution:** 3 good

**Summary:**

This paper addresses the problem of continuous category discovery, which is a natural extension of novel category discovery by sequentially feeding part of the targeting datasets into the model. The method builds on top of existing methods for novel category discovery and continual learning, namely, AutoNovel[6] and iCaRL[14], with a framework trained with grow and merge phases, containing two branches. Experiments on three public datasets have been presented showing obvious improvement over the compared methods.

**Questions:**

Please refer to the weaknesses above. In addition,

- Figure 4 and the illustration need to be clarified. The text is not well aligned with Fig 4, making the explanation rather unclear. For example, what are these selected clusters, what are those gray dots, which time-step is it, which are the old classes, and which are the new classes?
- As an additional comment to the weakness of the Experiments above, it would be more convincing to repeat on different random splits -- assuming the splits used in the experiments were randomly drawn, and also report the results on each split and each time step (more suitable for supplementary, but it would be useful to show how the method actually works in the intermediate steps)
- In Table 2 and 4, why DRNCD w/ LwF results are not reported, since it outperforms AutoNovel in NCD? But only the results of AutoNovel w/ LwF are reported.
- How was novelty detection threshold 0.6 determined?

**Limitations:**

Yes.

**Strengths And Weaknesses:**

Strengths:
+ The paper tacks the problem of continuous category discovery (NCD), which jointly considers the problem of novel category discovery and continual learning. This is a natural and interesting extension of NCD.
+ Good performance is achieved on three public datasets, outperforming the compared baselines.

Weaknesses:
- Though effective, the key components of the method are based on AutoNovel and iCaRL. The extra branches and grow-merge phases increased the complexity and meomery&computation cost for training.
- Experiments are only carried out on three relatively small datasets with a relatively small number of new classes in each time step. Also, the data splits for different time steps for the MI case vary a lot, which seems strange (as specified in the supplementary), why not adopt a consistent proportion for all? Or perhaps more different proportions can be used? How the different classes in each time step are constructed? randomly sampling or predefined in some way?
Only 10 classes are incremented at each time step, how about incrementing more classes? say 20, 50? For more challenging cases, a larger dataset could be more useful. It appears that more different datasets are used in the literature, e.g, CIFAR10, ImageNet,  Stanford-Cars FGVC-Aircraft in DRNCD [15].
- The new category number is assumed to be known in each incremental time step.

---

> ### Author Response · Authors · 2022-08-02
> **Response to Reviewer WTKt (3/3)**
>
> **[Concern 4]** About number of novel classes.
>
> We have conducted experiments without knowing the number of new categories. Estimating the number of novel classes can be easily employed in GMNet. Please see feedback in the summary of the paper revision for more details.
>
> **[Concern 5]** Figure 4 and the illustration need to be clarified.
>
> We sincerely apologize for the missing of description for Figure 4. For Figure (a), we compute the intra-class and inter-class distances for better understanding the growing and merging phases of GM. The results show that the classes tend to be more sparse during the growing phase for freely discovering novel categories, and then become tigher during the merging phase for better classification on all appeared classes. The quantitive results show consistentency with our motivation for GM. For Figure (b), the visualization results show a timestep of the whole GM framework, where the colored dots denote previously appeared classes, and the gray dots denote the newly appeared classes without labels at this timestep. After the growing phase with novel category discovery, the discovering process leads to sparser clusters for previous classes, which will decrease the classification performance on them. Then during the merging phase, we again cluster them tighter on both previous and new classes to maintain classification on currently appeared classes. The data comes from the experiments of GM model under CI scenario on the CIFAR-100 dataset, and the three subfigures in Figure (b) correspond to the feature distribution before the growing phase, after the growing phase, and after the merging phase at time step $t=3$.
>
> **[Concern 6]** DRNCD w/ LwF results are missing in Table 2 and 4.
>
> Thanks for the suggestion.
> The performance of DRNCD w/ LwF  in CI on all datasets, and MI/SMI on CIFAR-100 are provided as below, which are still inferior than GMNet.
>
> |                      | CIFAR-100       |                 | CUB-200         |                 | ImageNet-100    |                 | Stanford-Cars   |                 | FGVC-Aircraft   |                 |
> | -------------------- | --------------- | --------------- | --------------- | --------------- | --------------- | --------------- | --------------- | --------------- | --------------- | --------------- |
> |                      | $\mathcal{M}_f$ | $\mathcal{M}_d$ | $\mathcal{M}_f$ | $\mathcal{M}_d$ | $\mathcal{M}_f$ | $\mathcal{M}_d$ | $\mathcal{M}_f$ | $\mathcal{M}_d$ | $\mathcal{M}_f$ | $\mathcal{M}_d$ |
> | DRNCD (online) + LwF | 27.03           | 2.87            | 44.83           | 6.31            | 31.13           | 5.47            | 27.86           | 4.60            | 33.09           | 9.03            |
>
> |             | MI              |                 | SMI             |                 |
> | ----------- | --------------- | --------------- | --------------- | --------------- |
> |             | $\mathcal{M}_f$ | $\mathcal{M}_d$ | $\mathcal{M}_f$ | $\mathcal{M}_d$ |
> | DRNCD + LwF | 37.01           | 3.20            | 63.35           | 54.7            |
>
> **[Concern 7]** About novelty detection threshold.
>
> In our experiments, GM is not sensitive to this threshold and GM with the 0.6 theshhold will achieve relatively high performance. We provide the ablation study results on the novelty detection threshold in the revised version. The results show that though GM achieves the best performance with threshold 0.6, it maintains relatively high performance with the values in [0.4, 0.7] and consistently outperforms existing methods. The experiments are conducted on the CIFAR-100 dataset in MI scenario.
>
> | $\varepsilon$ | $\mathcal{M}_f$ | $\mathcal{M}_d$ |
> | ------------- | --------------- | --------------- |
> | 0.4           | 9.79            | 29.27           |
> | 0.5           | 10.29           | 32.33           |
> | 0.6           | 9.79            | 35.77           |
> | 0.7           | 10.16           | 30.37           |

---

> > ### Comment · Reviewer_WTKt · 2022-08-08
> > **Thanks for the response and further comments**
> >
> > I appreciate the authors' efforts in addressing my concerns, and some of my concerns have been properly addressed. There are a few remaining concerns as follows.
> >
> > 1. **Contributions**. The majority of techniques are borrowed from the literature, AutoNovel + iCaRL, the CE, BCE, and MSE losses. The PLL loss appears to be of no difference from the prototypical contrastive loss. Hence, I do feel the technical novelty of the paper is limited. Meanwhile, it is a good idea to have the filtering method to reject unreliable samples for training, but the Sifting approach appears to be not very effective (Table 6). Also, the experimented cases CI, DI, MI and SMI are plausible. The extended CCD setting is interesting and valuable. The introduced method is also effective, and the experimental results are also good. Besides, I noticed two concurrent works addressing the exact same problem from the recent ECCV. It would be good to have some discussion to benefit the readers, though it is reasonable to have no comparison at the moment. Note these two works do not affect my review of this paper as they are not published at the time when this paper is submitted.
> > [A] Novel Class Discovery without Forgetting, Joseph et al, ECCV 2022
> > [B] Class-incremental Novel Class Discovery, Roy et al, ECCV 2022
> >
> > 2. **Experimental results**.
> > GM is built on AutoNovel + iCaRL and actually contains the replay samples (the prototypes and exemplars) of previously seen classes, which is greatly helpful to avoid forgetting, while AutoNovel + LwF does not contain the replay samples, making the comparison a bit unfair for other baseline methods. Previous methods AutoNovel and DRNCD contain a labeled head to train on the labeled data. Has this been dropped in the comparison? If both of the above are true, it is not surprising that there is a huge gap on the performance of seen classes between the GM. Among CI, DI, MI and SMI, the latter two are the more practical and challenging. It is interesting to see DRNCD+LwF performing better than GM in the SMI case (Table 4).
> >
> > 3. **Evaluation metrics**.
> > It is good to measure the M_f and M_d to consider the ability to avoid forgetting and learn new classes. However, M_f is a relative measure while M_d is absolute measure. It would be also good to report the starting and ending performance in absolute terms (ie, classification accuracy) on the old classes to better reflect the actual performance on the old classes. Is it possible to evaluate the "old", "new", and "all" for each time step as commonly done in the literature of NCD/GCD, such that the performance will not mislead by the population of either "old" or "new" split? This is especially useful to measure the performance on MI and SMI cases.
> >
> > 4. **Threshold for novelty detection**.
> > Thanks for experimenting on CIFAR100 with different threshold values. However, my point was how to obtain a proper threshold for any given dataset. It is not valid to identify a value by checking the evaluation performance for real deployment. Also, intuitively, I feel the optimal value could be quite different for fine-grained datasets, while it seems the same threshold of 0.6 is used for fine-grained datasets?

---

> > > ### Author Response · Authors · 2022-08-09
> > > **Response to Reviewer WTKt (3/3)**
> > >
> > > **[Concern 3]**
> > >
> > > Here we provide detailed results, known class recognition and novel class discovery at each stage. For known class recognition, we also provide two indicators (one for known classes only, one for of all identified classes so far).
> > >
> > > | CI    | old (known classes only) | old (all identified classes so far) |  new |  all |
> > > | ----- | -----------------------: | ----------------------------------: | ---: | ---: |
> > > | $t=0$ |                     72.5 |                                72.5 |    - | 72.5 |
> > > | $t=1$ |                     68.3 |                                68.3 | 36.0 | 64.3 |
> > > | $t=2$ |                     65.5 |                                61.5 | 37.1 | 58.8 |
> > > | $t=3$ |                     62.6 |                                55.9 | 36.8 | 53.9 |
> > >
> > > | MI    | old (known classes only) | old (all identified classes so far) |  new |  all |
> > > | ----- | -----------------------: | ----------------------------------: | ---: | ---: |
> > > | $t=0$ |                     70.6 |                                70.6 |    - | 70.6 |
> > > | $t=1$ |                     66.9 |                                66.9 | 27.6 | 62.0 |
> > > | $t=2$ |                     63.5 |                                58.7 | 33.9 | 55.9 |
> > > | $t=3$ |                     60.7 |                                53.6 | 29.3 | 51.2 |
> > >
> > > | SMI   | old (known classes only) | old (all identified classes so far) |  new |  all |
> > > | ----- | -----------------------: | ----------------------------------: | ---: | ---: |
> > > | $t=0$ |                     70.6 |                                70.6 |    - | 70.6 |
> > > | $t=1$ |                     67.2 |                                67.2 | 32.1 | 62.8 |
> > > | $t=2$ |                     64.1 |                                59.8 | 38.4 | 57.4 |
> > > | $t=3$ |                     61.6 |                                55.3 | 34.6 | 53.2 |
> > >
> > > It can be seen that as the number of stages increases, the overall effect shows a smooth decline, but our method can still maintain high performance in the recognition of known classes, and achieve satisfactory novel class discovery performance.
> > >
> > >
> > > **[Concern 4]**
> > >
> > > We provide the results of the common fine-grained classification datasets, i.e., CUBS and Aircraft. The fine-grained recognition makes SMI and MI a great challenge, but GM can still surpass the comparison method and obtain stable performance when the value of threshold varies. The results indicate that we can select thresholds $\epsilon$ in a wide range (such as around the middle value of 0.5), even on different datasets, they will not have a great impact on the performance of GM. We can also find that GM outperforms other methods with the wide range of the threshold. Due to the time constraints, we provide experimental results under MI on CIFAR100, CUB, and Aircraft, and results under SMI on CIFAR100. The rest experimental results will be discussed in the final version.
> > >
> > > MI:
> > >
> > > |  | cub             |                 | aircraft        |                 | CIFAR100||
> > > | -------------------------------------- | --------------- | --------------- | --------------- | --------------- | --------------- | --------------- |
> > > |  | $\mathcal{M}_f$ | $\mathcal{M}_d$ | $\mathcal{M}_f$ | $\mathcal{M}_d$ | $\mathcal{M}_f$ | $\mathcal{M}_d$ |
> > > | GM ($\epsilon$=0.4)                                 | 18.97           | 20.97           | 10.80           | 24.79           | 9.79 |29.27|
> > > | GM ($\epsilon$=0.5)                                 | 23.65           | 24.49           | 8.20            | 26.32           | 10.29           | 32.33           |
> > > | GM ($\epsilon$=0.6)                                  | 21.52           | 22.15           | 8.48            | 27.30           |9.64            | 30.58      |
> > > | GM ($\epsilon$=0.7)                                | 18.99           | 19.89           | 8.02            | 26.70           |10.16           | 30.37           |
> > > | AutoNovel+LwF+replay                   | 43.79           | 15.45           | 43.38           | 9.80            |59.61|8.43|
> > >
> > > SMI:
> > >
> > > |  | CIFAR100 |      |
> > > | -------------------------------------- | -------- | ---- |
> > > |                                        |  $\mathcal{M}_f$        |   $\mathcal{M}_d$   |
> > > | GM ($\epsilon$=0.4) |          9.17|34.63|
> > > | GM ($\epsilon$=0.5)|         9.09|37.76|
> > > | GM ($\epsilon$=0.6) |         8.84|37.23|
> > > | GM ($\epsilon$=0.7) |        9.24|38.63|
> > > | AutoNovel+LwF+replay |     58.47     | 33.43     |
> > > | DRNCD+LwF+replay |       21.93   |  57.9    |

---

> > > ### Author Response · Authors · 2022-08-09
> > > **Response to Reviewer WTKt (2/3)**
> > >
> > > **[Concern 2]**
> > >
> > > (1) For the compared methods, i.e. AutoNovel and DRNCD, the labeled head trained on labeled data are already kept for comparison in the experiments. For the replay samples, we provide the performance of the refered methods with replay samples (which are the same with our GM) in the below:
> > >
> > > Experiments under MI:
> > >
> > > |                      | $\mathcal{M}_f$ | $\mathcal{M}_d$ |
> > > | -------------------- | --------------- | --------------- |
> > > | AutoNovel+LwF        | 61.89           | 5.59            |
> > > | DRNCD+LwF            | 35.44           | 4.10            |
> > > | AutoNovel+LwF+replay | 59.61           | 8.43            |
> > > | DRNCD+LwF+replay     | 34.83           | 2.30            |
> > > | GM                   | 9.65            | 30.58           |
> > >
> > > Experiments under SMI:
> > >
> > > |                      | $\mathcal{M}_f$ | $\mathcal{M}_d$ |
> > > | -------------------- | --------------- | --------------- |
> > > | AutoNovel+LwF        | 61.45           | 31.84           |
> > > | DRNCD+LwF            | 62.38           | 52.53           |
> > > | AutoNovel+LwF+replay | 58.47           | 33.43           |
> > > | DRNCD+LwF+replay     | 21.93           | 57.90           |
> > > | GM                   | 9.41            | 35.52           |
> > >
> > > We can find that AutoNovel and DRNCD benifit from the replay samples. However, it is still hard to tackle the challenges of CCD problem with only LwF and replay samples. We will include the results in the revised version.
> > >
> > > (2) The significant performance is not simply from the replay samples or combination of existing methods. GM is a generic framework, which includes grow and merge phases to solve the contradiction between discovering novel categories and maintaining the classification on known categories. It is shown in Tables 5 and 6 that our proposed method is critical for the $\mathcal{M}_f$ and $\mathcal{M}_d$ performance. Here we further provide the ablation studies in the challenging MI and SMI settings to demonstrate that the performance obtained by the proposed methods is not simply from a combination of existing methods.
> > >
> > > |                                                              | $\mathcal{M}_f$ | $\mathcal{M}_d$ |
> > > | ------------------------------------------------------------ | --------------- | --------------- |
> > > | (i) MI, GM w/o $\mathcal{L}_\text{SD}$                       | 22.04           | 22.48           |
> > > | (ii) MI, GM w/o $\mathcal{L}_\text{SD}$  + EMA               | 26.20           | 24.36           |
> > > | (iii) MI, GM w/o $\mathcal{L}_\text{SD}$  + EMA + $\mathcal{L}_\text{PLL}$ | 46.47           | 18.02           |
> > > | (iv) SMI, GM w/o $\mathcal{L}_\text{SD}$                     | 21.46           | 28.47           |
> > > | (v) SMI, GM w/o $\mathcal{L}_\text{SD}$  + EMA               | 26.51           | 31.67           |
> > > | (vi) SMI, GM w/o $\mathcal{L}_\text{SD}$  + EMA + $\mathcal{L}_\text{PLL}$ | 43.66           | 20.83           |
> > >
> > > We can find the same phenomenon under both MI and SMI, *i.e.*, $\mathcal{L}_\text{PLL}$ could decrease $\mathcal{M}_f$ significantly and also increase $\mathcal{M}_d$. EMA and $\mathcal{L}_\text{SD}$ could help the model leverage the effective representation learned from the initial stage, and further improve the performance. With the help of the proposed methods, GM obtains the best performance on both $\mathcal{M}_f$ and $\mathcal{M}_d$. It is interesting to see that  $\mathcal{M}_f$ is improved and $\mathcal{M}_d$ decreases after adding EMA to the model ((v) compared with (iv) and (ii) compared with (i)). We think maybe that data from known categories contained in MI and SMI scenario enhances the representation of the known classes of the dynamic branch, while EMA improves the representation of the known classes as well, which leads to lower forgetting with a slight decrease in the discovery ability.
> > >
> > > (3) For the performance of DRNCD + LwF in Table 4, although it achieves a good $\mathcal{M}_d$ performance, it performs very poorly on $\mathcal{M}_f$ and almost forgets the known classes recognition capabilities, which is contrary to the starting point of the CCD problem. GM can well handle both the two tasks of known class recognition and novel class discovery. Experiments on multiple settings (CI, DI, MI and SMI) of multiple datasets show the superiority of GM compared to other methods.

---

> > > ### Author Response · Authors · 2022-08-09
> > > **Response to Reviewer WTKt (1/3)**
> > >
> > > Thanks for your feedback. We are glad to hear that some of your concerns have been properly addressed. Here we provide further analysis of the remaining concerns. We hope the answer could address all your concerns. We thank you again for your timely follow-ups and look forward to your reply.
> > >
> > > **[Concern 1]**
> > >
> > > (1) About the technical novelty.
> > >
> > > We sincerely clarify that the simple combination of existing techniques, e.g. AutoNovel, iCaRL, BCE loss, could not effectively address the Continuous Category Discovery (CCD) problem. The main challenge of CCD is the contradiction between discovering novel categories and maintaining the classification on known categories. The settings of CCD also require the continuous merging of the known category set and corresponding models.
> > >
> > > Therefore, the technical novelty of this work is to propose a grow and merge (GM) framework for CCD, instead of the detailed design of how to discover novel categories. In the GM framework, we also propose key modules, i.e., static-dynamic branch, and further design static-dynamic distillation loss and PLL based on this, they significantly improve the final performance. The static-dynamic distillation loss takes advantage of the different features of the proposed static and dynamic branches to improve the feature representation. PLL employs the exemplar that is contiguously constructed in the CCD stages and so can benefit from both labeled data and sample data identified in CCD stages to improve the representation, which is different from PCL which employs contrastive learning and clustering in unsupervised learning.
> > >
> > > Existing approaches for novel category discovery could be directly plugged into the growing phase of our GM framework. We also noticed that the same MSE and BCE losses from AutoNovel are used in the concurrent work [B], which demonstrates the simple usage of existing techniques could not address the problem.
> > >
> > > (2) About the effectiveness of Sifting.
> > >
> > > The sifting approach aims to filter out the potential bad cases of novel category clustering, which mainly benefits the final discovery accuracy (+2.3%). The sifting in GM works as an improving technique for better representation of new classes and is demonstrated to contribute to better performance of novel category discovery.
> > >
> > > (3) Comparison with concurrent works.
> > >
> > > Though published after the submission of our work, we present a brief comparison with the referred concurrent works (reference [A] and [B]) for better understanding.
> > >   - **Problems**: The main difference is that our studied CCD is MULTI-STAGE with more than one batch of novel category samples contiguously confronted, while the settings of [A] and [B] are SINGLE-STAGE with all the novel categories coming at once. Such a difference of settings leads to more challenges, such as avoiding forgetting of previously discovered novel categories, the continual merging of discovered categories into the known set, identifying both labeled and previously discovered categories, etc. Our GM framework could address the above problems under the more complex multi-stage settings.
> > >   - **Experimental scenarios**: We consider four different scenarios where the coming data stream may contain labeled or unlabeled, known or novel categories. The number of novel categories is also considered to be unknown in the revised version. For [A] and [B], they only consider the coming data to contain purely novel categories with a known number of categories.
> > >   - **Evaluation**: Both [A] and [B] report the final accuracy on old, new, and all categories, without reporting the initial accuracy on old categories to indicate forgetting. We design a more detailed metric to indicate the forgetting of both labeled and discovered categories for better comparison. The absolute values are reported in [Concern 3].
> > >   - **Method**: We all maintain a frozen backbone pre-trained on the initial labeled data and guide the following training with feature distillation. The main differences between our proposed GM and [A, B] are that GM handles classification and discovery in different phases with growing and merging, which can resolve the features contradiction. While in [A, B], these different targets are optimized at the same time.
> > >
> > >
> > > References:
> > >
> > > [A] Novel Class Discovery without Forgetting, Joseph et al, ECCV 2022.
> > >
> > > [B] Class-incremental Novel Class Discovery, Roy et al, ECCV 2022.
> > >
> > > [C] A Unified Objective for Novel Class Discovery. Fini et al, ICCV 2021.

---

> ### Author Response · Authors · 2022-08-02
> **Response to Reviewer WTKt (2/3)**
>
> **[Concern 3]** About the suggested experiments: We have conducted the experiments as suggested.
>
> (1)  Different data splits:
>
> Here we provide results on CIFAR-100 dataset in MI scenario under 3 times random split.
>
> |                           | $\mathcal{M}_f$ | $\mathcal{M}_d$ |
> | ------------------------- | --------------- | --------------- |
> | AutoNovel                 | 63.25$\pm$ 0.19 | 5.74$\pm$0.17   |
> | AutoNovel + LwF           | 61.89$\pm$0.87  | 5.59$\pm$1.99   |
> | DRNCD                     | 43.81$\pm$1.71  | 5.12$\pm$0.97   |
> | GM w/o. novelty detection | 4.74$\pm$0.87   | 10.61$\pm$3.37  |
> | GM                        | 9.65$\pm$0.32   | 30.58$\pm$1.13  |
>
> Here we provide GM performance at each time step on CIFAR-100 dataset in MI scenario under 3 times random split.
>
> | $t$  | $\mathcal{M}_f$ | $\mathcal{M}_d$ |
> | ---- | --------------- | --------------- |
> | 2    | 3.97$\pm$0.06   | 33.67$\pm$4.95  |
> | 3    | 6.70$\pm$0.18   | 36.60$\pm$0.87  |
> | 4    | 9.86$\pm$0.33   | 36.36$\pm$0.81  |
>
> The results are consistent with the original ones. In final version, we will update all the results with the mean and std under 3 random splits.
>
> (2) About more incremental classes at each time step:
>
> For the data split in MI setting, because the later stages involve more categories, in order to maintain a consistent amount of incremental samples, the proportion of data from different categories in the contiguous stages will generally decline, and because of the number of new categories added in each stage is also different, so the proportion of data sampled at each stage will also fluctuate sightly.
> We provide experimental results of 20 incremental classes at each time step on CI scenario and CIFAR-100 dataset to investigate the robustness of GMNet to number of incremental classes, where GMNet can be well compatible with different number of incremental classes and obtain consistent performance improvement.
>
> |                                                    | $\mathcal{M}_f$ | $\mathcal{M}_d$ |
> | -------------------------------------------------- | --------------- | --------------- |
> | lower-bound of $\mathcal{M}_f$ (offline K-Means)   | 6.73            | 16.53           |
> | upper-bould of $\mathcal{M}_d$ (offline AutoNovel) | 5.87            | 31.57           |
> | AutoNovel (online)                                 | 62.59           | 28.8            |
> | DRNCD (online)                                     | 67.05           | 6.58            |
> | AutoNovel (online) + LwF                           | 15.04           | 27.15           |
> | DRNCD (online) + LwF                               | 67.17           | 5.87            |
> | iCaRL (fixed exemplars) + LwF                      | 30.52           | -               |
> | GM (Ours)                                          | 10.20           | 35.73           |
>
> (3) About more datasets.
>
> We provide results on Stanford-Cars and FGVC-Aircraft under CI scenario here, where GMNet still outperform all other compared methods.
>
>  Stanford-Cars:
>
> |                                                    | $\mathcal{M}_f$ | $\mathcal{M}_d$ |
> | -------------------------------------------------- | --------------- | --------------- |
> | lower-bound of $\mathcal{M}_f$ (offline K-Means)   | 2.53            | 10.05           |
> | upper-bound of $\mathcal{M}_d$ (offline AutoNovel) | 11.27           | 19.46           |
> | AutoNovel (online)                                 | 71.88           | 16.46           |
> | DRNCD (online)                                     | 54.50           | 1.83            |
> | AutoNovel (online) + LwF                           | 71.60           | 20.35           |
> | DRNCD (online) + LwF                               | 27.86           | 4.6             |
> | iCaRL (fixed exemplars) + LwF                      | 24.71           | -               |
> | GM (Ours)                                          | 18.00           | 25.16           |
>
> FGVC-Aircraft:
>
> |                                                    | $\mathcal{M}_f$ | $\mathcal{M}_d$ |
> | -------------------------------------------------- | --------------- | --------------- |
> | lower-bound of $\mathcal{M}_f$ (offline K-Means)   | 3.73            | 14.70           |
> | upper-bound of $\mathcal{M}_d$ (offline AutoNovel) | 4.85            | 26.90           |
> | AutoNovel (online)                                 | 53.84           | 25.23           |
> | DRNCD (online)                                     | 43.63           | 2.21            |
> | AutoNovel (online) + LwF                           | 44.90           | 11.67           |
> | DRNCD (online) + LwF                               | 33.09           | 9.03            |
> | iCaRL (fixed exemplars) + LwF                      | 21.31           | 6.87            |
> | GM (Ours)                                          | 8.41            | 30.82           |
>
> Due to the tight schedule during the rebuttal, we will include results of these new datasets and imagenet in the final version.

---

> ### Author Response · Authors · 2022-08-02
> **Response to Reviewer WTKt (1/3)**
>
> We thank the reviewer for the careful reading and valuable feedback to our submission. Here we provide more experimental results and detailed description to demonstrate the effectiveness of GM. We hope our responce could solve your concerns.
>
> **[Concern 1]** The key components of the method are based on AutoNovel and iCaRL.
>
> We sincerely apologize for the unclear presentation of our method, which has been polished in the revised version. One of the main contributions of this paper is to find that there exists a contradiction between existing continual classification method (e.g. iCaRL) and category discovery method (e.g. AutoNovel), as shown in Figure 2. Thus, a simple combination of these methods could not achieve satisfying performance as shown in Table 2. The proposed GM framework addresses this problem via seperating this two tasks in a growing and merging phases, respectively. Then typical methods (e.g. AutoNovel, iCaRL) could be freely plugged into our GM framework for this challenge. We present more discussion in the revised version.
>
> **[Concern 2]** The extra branches increased the meomery & computation cost for training.
>
> The main extra computation & memory cost are introduced by the static branch. Such a two-branch architecture is a typical design in various fields of machine learning, such as Teacher-Student Network[1], LwF[2], SimSiam[3], etc. Furthermore, the static branch is frozen during the merging phase without introducing additional computation for backward propagation (BP).
>
> [1] Harnessing deep neural networks with logic rules, ACL 2016.
>
> [2] Learning without Forgetting, TPAMI 2017.
>
> [3] Exploring Simple Siamese Representation Learning, CVPR 2021.

---

> ### Author Response · Authors · 2022-08-06
> **A Gentle Reminder of Feedbacks**
>
> Dear Reviewer WTKt ,
>
> Thanks again for your careful reading and valuable comments to improve our submission. We want to leave a gentle reminder due to the closing end time of the discussion period. We have tried to address all your concerns with detailed explanations and results, and revised the paper correspondingly. We would really appreciate feedback to make sure the responses and revisions have addressed all your concerns, or whether there is any leftover concern we can address.
>
> Sincerely
> Authors of Paper5057

---

### Official Review · Reviewer_kWya · 2022-07-09

**Rating:** 6
**Confidence:** 5
**Soundness:** 3 good
**Presentation:** 3 good
**Contribution:** 2 fair

**Summary:**

This paper advances the task of novel category discovery to a dynamic setting where unlabelled data are continuously fed into a model for category discovery, refered as continuous category discovery.
The paper also proposed a grow and merge framework to solve this problem by balancing between novel category discovery and merging the newly discovered information with the knowledge already know to the model.


**Questions:**

Q1: In fig. 2, the paper aim to show that the features from the known categories and the novel categories are different, however, this visualization is not quantitative and could cause a strong bias if we only visualize a small subset of the dataset. Although I think the overall observation is correct, I would suggest to use the notion like uniformity and alignment [R1] to better show the main motivation of the paper.

Q2: In the sample sifting part, I did not quite understand the reasoning of find the maximum distance between a data point and the j nearest neighbors, isn't the maximum distance between a data point and its top-j neighbors always at the j-th nearest neighbor? In other word, is this measure always depends on the parameter j? I am also wonderring how about using a averaged distance between the data point and all its top-j nearest neighbor.

Q3: The setting assumes that the number of categories at time t is known, however this is an unrealistic assumption, because the novel categories are unknown, how can we assume we know the number of categories? In previous works on novel category discovery, there are methods that can be used to estimate the number of novel categories [R2], and so it makes sense for NCD methods to assume the category is known, however in the CCD case proposed in this paper, I would suggest the paper to include discussion of how to apply the algorithm in [R2] to estimate the number of categories in each time step in the CCD setting, and also include experiments of evaluate the proposed method using the estimated category number.

Q4: The method part of this paper is quite hard to understand, because it involves concepts that are not properly defined (such as exemplars and prototypes, the paper only states it use these terms as in iCaRL makes the paper not self-contained), and the framework seems to be quite complex, I would suggest to make it clear about every concept mentioned in the paper, and perhaps also include a method diagram to better demonstrate the idea of the framework.

Q5: The framework is quite complex, involving four losses, how to balance between these losses? The paper simply add them together without weighting, it would be great if we can have a study about the weighting between the four losses.

Q6: The experiments in Table 4 seem weird, the DRNCD method performed lowest on the MI setting, but the performance on SMI setting improves a lot, this is strange, could this be a sign that the setting are not stable, so that the variance of the experiments is large? Also, I would appreciate it if the paper could provide error bars on the experiments results.


Minor:

1. The paper is not very self-contained, the term exemplars and the initialization of exemplars and prototypes are used but not defined, the paper directly cites iCaRL for this, IMO, a good paper should be self-contained, at least the definition of these terms should be in the main text.
2. Typo at line 266 phrase -> phase


[R1] Understanding Contrastive Representation Learning through Alignment and Uniformity on the Hypersphere, ICML 2020

[R2] Learning to Discover Novel Visual Categories via Deep Transfer Clustering, ICCV 2019


**Limitations:**

The most noticable limitation of the proposed method and setting is that the number of novel categories is assumed to be konwn, I would suggest the author to try to include a discussion on this and try to expand the algorithm in [R2] for the proposed setting.

**Strengths And Weaknesses:**

S1: The proposed experimental setting is properly defined, and could be of interest for the novel category discovery community.

S2: The proposed grow and merge framework seems to perform well on the proposed experimental setting.

W1: The overall pipeline is complex, however I think the evaluation and discussion of the results are not enough, see Questions.

W2: The implicit assumption that the number of categories is known seems to be unrealistic.

Overall, I think this paper has its merit, but the defect outweight its merit, and the presentation of the work needs to be improved.

---

> ### Author Response · Authors · 2022-08-02
> **Response to Reviewer kWya (2/2)**
>
>
> **[Concern 4]** About presentation: We have carefully re-writen this section with more clear definition of concepts (self-contained in this paper), emphasizing the key components with better description and hiddening some less important details (included in the supplementary materials). We marked the modified part in the revised version with color for better reading.
>
> **[Concern 5]** About the loss weights: GM is a general framework that solves the CCD problem through the grow and merge processes. On this basis, we further enhance the performance through components such as EMA, Sifting and PPL. In the abalation study, removing these components, GM can still achieve better performance than other methods. For the influence of loss weights, we vary the weight of one loss and fix the weights of other losses to 1. The experiments are conducted on CI scenario in CIFAR-100 dataset:
>
> Vary weight of $\mathcal{L}_\text{BCE}$:
>
> | Weight | $\mathcal{M}_f$ | $\mathcal{M}_d$ |
> | ------ | --------------- | --------------- |
> | 0.1    | 9.61            | 21.00           |
> | 0.5    | 9.41            | 33.77           |
> | 1.0    | 9.79            | 35.77           |
> | 5.0    | 9.84            | 36.6            |
> | 10.0   | 10.07           | 36.3            |
>
> Vary weight of $\mathcal{L}_\text{MSE}$:
>
> | Weight | $\mathcal{M}_f$ | $\mathcal{M}_d$ |
> | ------ | --------------- | --------------- |
> | 0.1    | 10.21           | 32.47           |
> | 0.5    | 9.81            | 35.07           |
> | 1.0    | 9.79            | 35.77           |
> | 5.0    | 9.10            | 35.57           |
> | 10.0   | 8.44            | 31.63           |
>
> Vary weight of $\mathcal{L}_\text{PLL}$:
>
> | Weight | $\mathcal{M}_f$ | $\mathcal{M}_d$ |
> | ------ | --------------- | --------------- |
> | 0.1    | 9.11            | 37.03           |
> | 0.5    | 9.45            | 36.23           |
> | 1.0    | 9.79            | 35.77           |
> | 5.0    | 30.59           | 35.13           |
> | 10.0   | 71.07           | 0.66            |
>
> Vary weight of $\mathcal{L}_\text{SD}$:
>
> | Weight | $\mathcal{M}_f$ | $\mathcal{M}_d$ |
> | ------ | --------------- | --------------- |
> | 0.1    | 10.20           | 36.67           |
> | 0.5    | 10.41           | 35.93           |
> | 1.0    | 9.79            | 35.77           |
> | 5.0    | 9.81            | 33.00           |
> | 10.0   | 10.01           | 32.70           |
>
> We can find that although 4 losses are involved, our method shows robust to different loss weights. Except for using too small weight for $\mathcal{L}_\text{BCE}$ and too large weight for $\mathcal{L}_\text{PLL}$, in other cases, our model can obtain superior performance with a large range of loss weight (from 0.1 to 10).
>
> **[Concern 6]** About the performance of DRNCD
>
> Compared to MI setting, the continuous data in SMI include labeled data, which enable existing methods (e.g. DRNCD) easier to learn effective features representation and obtaining better performance on $\mathcal{M}_d$. Here we report the mean and std results on 3 random splits for DRNCD in CIFAR-100 dataset in the Table 4, which obtain performance close to the original ones. In final version, we will update all the experimental results with the mean and std under 3 random splits.
>
> |      | $\mathcal{M}_f$ | $\mathcal{M}_d$ |
> | ---- | --------------- | --------------- |
> | MI   | 43.81$\pm$1.71  | 5.12$\pm$0.98   |
> | SMI  | 56.27$\pm$1.13  | 47.63$\pm$1.66  |

---

> > ### Comment · Reviewer_kWya · 2022-08-04
> > **Thanks for the rebuttal and addition results**
> >
> > [Concern 1] Thanks for providing the additional results, but the results seem to be strange to me as the difference in the absolute value is small, leading to a question: is this significant?
> > I have also checked the fig.4 in the paper and noticed that the value range is also very small, maybe plotting the curve and uniformity and alignment with an error bar can better illustrate this idea.
> >
> > [Concern 2] This clarification is clear, this makes sense to me now.
> >
> > [Concern 3] Thanks for the results, it looks good to me. Please consider adding this to the final version of the paper.
> >
> > [Concern 4] Thanks for revising the paper, now it looks clear to me, although I would suggest to include every loss definition in the main paper, right now MSE loss is missing.
> >
> > [Concern 5] Thanks for the results. I have two additional questions about the results, 1) the weight range from 0.1 to 10.0, is this to wide? 2) For the MSE loss, common practice is to a ramp-up function as the weight for MSE loss, such as AutoNovel, but it seems that in these experiments, the weight is fixed? Also I would suggest to include MSE loss in the main paper, now it is missing.
> >
> > [Concern 6] Thanks for the results, it is surprising that the performance on M_d is so large between MI and SMI, yet the proposed method could still achieve good results on M_d under MI, while other methods fail at M_d, I would suggest to add a bit more discussion on this.
> >
> >
> >
> > Minor:
> >
> > Typo in the revised paper,
> > Table 4. DRNCE -> DRNCD

---

> > > ### Author Response · Authors · 2022-08-06
> > > **Response to Reviewer kWya (3/3)**
> > >
> > > **[Concern 6]** Discussion about the effectiveness of GMNet on $\mathcal{M}_d$
> > > For the compared methods, they can hardly solve the contradiction between the two tasks of classification and novel classes discovery in contiguous stages well, leading to the degraded performance . Labeled data is introduced in SMI, so that the model can fit the distribution of new data and learn the  novel category information at each stage, reducing the difficulty of  novelty detection and improving the performance of $\mathcal{M}_d$, but even so, existing methods still not well compatible with the two tasks, led to a more severe forgetting effect.
> > > For GMNet, through the static-dynamic branch and exemplar structure and $\mathcal{L}_\text{SD}$ and $\mathcal{L}_\text{PLL}$ on top of this, the model can be well compatible with two tasks in contiguous stages. we provide ablation study for them (on CIFAR-100, MI scenario):
> > >
> > > |                                                              | $\mathcal{M}_f$ | $\mathcal{M}_d$ |
> > > | ------------------------------------------------------------ | --------------- | --------------- |
> > > | GM                                                           | 9.83            | 32.40           |
> > > | GM w/o. $\mathcal{L}_\text{SD}$                              | 22.23           | 22.27           |
> > > | GM w/o. $\mathcal{L}_\text{SD}$, EMA                         | 25.94           | 24.50           |
> > > | GM w/o. $\mathcal{L}_\text{SD}$, EMA, $\mathcal{L}_\text{PLL}$ | 45.60           | 19.93           |
> > >
> > > We can find that, $\mathcal{L}_\text{PLL}$ could decrease $\mathcal{M}_f$ significantly and also increase $\mathcal{M}_d$. EMA and $\mathcal{L}_\text{SD}$ could help the model leverage the effective representation learned from the initial stage, and further improve the performance. With the help of proposed methods, GM obtains the best performance on both $\mathcal{M}_f$ and $\mathcal{M}_d$.
> > >
> > >
> > > Thanks again for your sincere suggestions and your timely follow-ups. Please let us know if there are leftover concerns and we would be glad to do our utmost to address them.

---

> > > > ### Comment · Reviewer_kWya · 2022-08-06
> > > > **Thanks for the results**
> > > >
> > > > Thanks for the additional results.
> > > > My concerns are mostly resolved, so I raised my score to 6 weak accept.

---

> > > > > ### Author Response · Authors · 2022-08-06
> > > > > **Thanks**
> > > > >
> > > > > We are encouraged that you raised the score and glad that your concerns have mostly been addressed. Please let us know if there is any other point you would want to further discuss to finish addressing your concerns.

---

> > > ### Author Response · Authors · 2022-08-06
> > > **Response to Reviewer kWya (2/3)**
> > >
> > > **[Concern 5]** About the range of loss weight analysis and MSE loss.
> > > For the former, we add denser loss weight values for experiments (make the current values include 0.1, 0.25, 0.5, 0.75, 1.0, 2.5, 5, 7.5 and 10) and provide experimental results in below. As we can see in these results, our method still achieves the same excellent performance as before.
> > > For the latter, we also conducted experiments as below.
> > >
> > > | $\mathcal{L}_\text{BCE}$ Weight | $\mathcal{M}_f$ | $\mathcal{M}_d$ | $\mathcal{L}_\text{MSE}$ Weight | $\mathcal{M}_f$ | $\mathcal{M}_d$ | $\mathcal{L}_\text{PLL}$ Weight | $\mathcal{M}_f$ | $\mathcal{M}_d$ | $\mathcal{L}_\text{SD}$ Weight | $\mathcal{M}_f$ | $\mathcal{M}_d$ |
> > > | ------------------------------- | --------------- | --------------- | ------------------------------- | --------------- | --------------- | ------------------------------- | --------------- | --------------- | ------------------------------ | --------------- | --------------- |
> > > | 0.1                             | 9.61            | 21.00           | 0.1                             | 10.21           | 32.47           | 0.1                             | 9.11            | 37.03           | 0.1                            | 10.20           | 36.67           |
> > > | 0.25                            | 8.94            | 27.93           | 0.25                            | 10.61           | 34.63           | 0.25                            | 9.84            | 37.17           | 0.25                           | 10.29           | 37.13           |
> > > | 0.5                             | 9.41            | 33.77           | 0.5                             | 9.81            | 35.07           | 0.5                             | 9.45            | 36.23           | 0.5                            | 10.41           | 35.93           |
> > > | 0.75                            | 10.36           | 35.17           | 0.75                            | 10.11           | 35.80           | 0.75                            | 9.94            | 35.87           | 0.75                           | 10.49           | 36.9            |
> > > | 1.0                             | 9.79            | 35.77           | 1.0                             | 9.79            | 35.77           | 1.0                             | 9.79            | 35.77           | 1.0                            | 9.79            | 35.77           |
> > > | 2.5                             | 9.94            | 37.00           | 2.5                             | 10.37           | 39.87           | 2.5                             | 20.43           | 35.23           | 2.5                            | 10.49           | 36.9            |
> > > | 5.0                             | 9.84            | 36.6            | 5.0                             | 9.10            | 35.57           | 5.0                             | 30.59           | 35.13           | 5.0                            | 9.81            | 33.00           |
> > > | 7.5                             | 10.27           | 33.33           | 7.5                             | 9.72            | 34.2            | 7.5                             | 71.07           | 1.23            | 7.5                            | 9.87            | 33.4            |
> > > | 10.0                            | 10.07           | 36.3            | 10.0                            | 8.44            | 31.63           | 10.0                            | 71.07           | 0.66            | 10.0                           | 10.01           | 32.70           |
> > >
> > > We also conduct experiment of the ramp-up function as the weight of MSE loss, as shown below. The model with ramp-up weight performs slightly better on $\mathcal{M}_f$ and worse on $\mathcal{M}_d$.
> > >
> > > |                                                    | $\mathcal{M}_f$ | $\mathcal{M}_d$ |
> > > | -------------------------------------------------- | --------------- | --------------- |
> > > | GM                                                 | 9.79            | 35.77           |
> > > | GM with ramp-up weight of $\mathcal{L}_\text{MSE}$ | 8.80            | 34.07           |

---

> > > ### Author Response · Authors · 2022-08-06
> > > **Response to Reviewer kWya (1/3)**
> > >
> > > Thanks for your feedback. We are glad to hear that some of your concerns have been addressed by our explanations.
> > >
> > > With respect to your suggestions, we do the following editing (We put most of them temporarily in the Supplementary Material, but we will squeeze part of them to the main paper if it is accepted because of the additional one page.):
> > >
> > > - We supplement the error bar of the alignment and uniformity in the supplementary material, and update the Figure 4. (a) with error bar in the main paper.
> > > - We add the experimental results about estimating the number of  novel categories in the supplementary material (We will move them to the main paper if accepted.)
> > > - We add the definition of the $\mathcal{L}_\text{MSE}$ in the main paper.
> > > - We add experimental results about the dense loss weight and the ramp-up function in the supplementary material.
> > > - We add discussion about the effectiveness of GMNet on $\mathcal{M}_d$ in supplementary material.
> > >
> > > Here we provide further analysis for the remaining concerns:
> > >
> > > **[Concern 1]** About the error bar of the uniformity and alignment.
> > > Thanks for your comments, we perform err bar calculations for uniformity and alignment, and we can see that the std of these values is very small, so it can reflect the significance of the results and our motivation is reasonable.
> > >
> > > | Alignment        | model A          | model B          | GM model         |
> > > | ---------------- | ---------------- | ---------------- | ---------------- |
> > > | Known Categories | -1.008$\pm$0.007 | -1.153$\pm$0.003 | -0.825$\pm$0.007 |
> > > | Novel Categories | -1.118$\pm$0.005 | -1.009$\pm$0.008 | -0.923$\pm$0.009 |
> > >
> > > | Uniformity       | model A          | model B          | GM model         |
> > > | ---------------- | ---------------- | ---------------- | ---------------- |
> > > | Known Categories | -2.704$\pm$0.008 | -2.625$\pm$0.005 | -2.510$\pm$0.012 |
> > > | Novel Categories | -2.631$\pm$0.009 | -2.754$\pm$0.008 | -2.397$\pm$0.012 |
> > >
> > > Due to the sparsity of high-dimensional space, the absolute value of the distance between features tends to be smaller[1]. The results of error bar shows that our observation is statistically significant, where the std. are smaller than the values of metric with consistence, which is consistent with our hypothesis. With the help of TSNE and the visualization in Figure 2, we can find that the difference of feature distributions of different models is obvious.
> > >
> > > The curve in Firgure 4 has the same phenomenon. The error bar in Figure 4. is updated, and we also provide the numerical values below.
> > >
> > > |                | Before growing $t=2$ | Before merging $t=2$ | After merging $t=2$ | Before merging $t=3$ | After merging $t=3$ | Before merging $t=4$ | After merging $t=4$ |
> > > | -------------- | -------------------- | -------------------- | ------------------- | -------------------- | ------------------- | -------------------- | ------------------- |
> > > | Inner-distance | 0.520$\pm$0.003      | 0.498$\pm$0.004      | 0.480$\pm$0.006     | 0.501$\pm$0.008      | 0.476$\pm$0.005     | 0.491$\pm$0.008      | 0.467$\pm$0.004     |
> > > | Intra-distance | 1.524$\pm$0.005      | 1.458$\pm$0.006      | 1.474$\pm$0.004     | 1.429$\pm$0.006      | 1.484$\pm$0.008     | 1.443$\pm$0.008      | 1.464$\pm$0.005     |
> > >
> > > [1] When is “nearest neighbor” meaningful?, International Conference on Database Theory. Springer, Berlin, Heidelberg, 1999.
> > >
> > > **[Concern 3&4]** We add the experimental results about estimating the number of  novel categories in the supplementary material (We will move them to the main paper if accepted.), and the definition of the $\mathcal{L}_\text{MSE}$ in the main paper.

---

> ### Author Response · Authors · 2022-08-02
> **Response to Reviewer kWya (1/2)**
>
> We thank the reviewer for the careful reading and valuable feedback to our submission. We are encouraged that the reviewer found our proposed setting could be of interest for the novel category discovery community. We sincerely apologize for the unclear presentation. Here we provide more experimental results and detailed description to demonstrate the effectiveness of GM. We hope our responce could solve your concerns.
>
>
> **[Concern 1]** About the distance in Fig. 2: The uniformity and alignment (on CIFAR-100 dataset in CI scenario) are calculated and provided as follows:
>
> | Alignment        | model A | model B | GM model |
> | ---------------- | ------- | ------- | -------- |
> | Known Categories | -1.01   | -1.15   | -0.82    |
> | Novel Categories | -1.11   | -1.00   | -0.91    |
>
> | Uniformity       | model A | model B | GM model |
> | ---------------- | ------- | ------- | -------- |
> | Known Categories | -2.71   | -2.63   | -2.50    |
> | Novel Categories | -2.62   | -2.74   | -2.39    |
>
> The results show that the changes of these two indicators are consistent with our assumption. Classifcation model (A) show smaller uniformity and larger alignment on known categories than novel categories and novelty detection model show the opposite results, which indicates feature discriminative ability of the two model focus on different sample types (novel or known categories). Compared to the two models, GM model show better uniformity and alignment on both novel and known categories. Moreover, in Fig. 4 of the submission we provide inter-/intra- class distance changes during the GM training, which further suggest the effectiveness of our method.
>
> **[Concern 2]** About alternatives of sample sifting distance: We assume that if the data point has j highly similar samples (< given threshold), it can be considered that it belongs to a potential new category, and should not be considered as noise, which means the distance between it and its j-th nearest neighbor are smaller than given threshold. The influence of different j is investigated on CI scenario and CIFAR-100 dataset:
>
> | j    | $\mathcal{M}_f$ | $\mathcal{M}_d$ |
> | ---- | --------------- | --------------- |
> | 5    | 10.06           | 34.10           |
> | 10   | 10.11           | 35.03           |
> | 15   | 9.79            | 35.77           |
> | 20   | 9.64            | 34.13           |
> | 25   | 9.54            | 33.87           |
>
> Different j will cause slight fluctuations of model performance, but excellent performance can still be achieved by GMNet.
>
> The results of average distance of top-j nearest neighbor in CI scenario and CIFAR-100 dataset are shown as below:
>
> |              | $\mathcal{M}_f$ | $\mathcal{M}_d$ |
> | ------------ | --------------- | --------------- |
> | GM           | 9.79            | 35.77           |
> | avg. sifting | 10.40           | 33.33           |
>
> Average distance performs slightly worse than maximum distance of top-j NN.
>
> **[Concern 3]** About number of novel classes: We have conducted experiments without knowing the number of new categories. Estimating the number of novel classes can be easily employed in GMNet. Please see feedback in the summary of the paper revision for more details.

---

### Official Review · Reviewer_oK8E · 2022-07-18

**Rating:** 6
**Confidence:** 4
**Soundness:** 3 good
**Presentation:** 2 fair
**Contribution:** 3 good

**Summary:**

Authors propose a continuous setting of GCD called CCD where unlabeled data from novel and known categories appear in batches. And they also propose a framework/method to iteratively update representation & novel category discovery for CCD. In proposed framework, authors first perform novelty detection and discover novel categories & update representation on batch of unlabeled data via pairwise similarity of representation & regularize via static branch. Then based on estimate of current cluster assignments, they remove outliers of clusters to update exemplars and retrain representation with pseudo labels to merge categories, and finally merge static and dynamic representation with a moving avg. strategy.


**Questions:**

In line 60 of page 2, 'we maintain a small set of training examples for each category for both known & discovered'. Can you comment and clarify size of this small dataset stored and all the steps where this dataset is leveraged, so is this small dataset not used for merging and only in growing as stated in line 62?

**Limitations:**

Currently framework is adopted to a certain choice of self-supervised representational learning, though framework is generic its effectiveness with other techniques is to be evaluated empirically.
And as authors pointed out this framework assumes no.of.novel categories in each time-step a priori but could be addressed in future work.

**Strengths And Weaknesses:**

Key contributions of paper is Proposal of CCD which useful continuous adaptation variation of GCD. And they propose a framework to address the setup and alleviate challenges of catastrophic forgetting, noisy gradient information available in CCD unlike GCD.

Strengths:
	Addresses an important problem of interest of practical value.
	Proposed framework could potentially be applied with various techniques.
	Multiple settings of experiments for evaluations and good analysis in experiments.

Weakness
	Lack of evaluation with different techniques, especially considering a continuous version of GCD or other semi/self-supervised representation updates.
    Lack of clarity in writing few details w.r.t setting and experiments, especially what and how if any additional data is used beyond exemplars. Also it is mentioned as limitation that number of novel classes at any time is known a-priori in supplementary, to make it clear it should be stated in main paper too while defining $L_{BCE}$ and in the setting.

Suggestion to Authors:
    Proposed grow and merge framework is a useful generic solution and is potentially agnostic to techniques used for self-supervised representational learning and to updating representation within CCD iterations.
    To decouple effectiveness of proposed framework and get better information w.r.t dependency on exact techniques especially w.r.t online vs offline loss in performance, it could be fruitful to adopt proposed framework to some other method too?

E.g. [1] for GCD, a follow up to AutoNovel adopted in this paper or any other method for GCD could be another upper-bound to consider? This is a suggestive example, authors may choose other method of representation learning and updates to adopt proposed framework.
		For [1] may be worthwhile to consider performing an experiment without WTA in growing phase and use techniques from GCD, with and without appropriate $L_{SD}$.
		Where for an unlabeled dataset , update representation with self-supervised instance contrastive learning and also supervised contrastive learning leveraging small dataset?, and perform semi-supervised k-means++ applied by GCD for clustering.
	        On this representation, get final performance with and without retraining/additional fine-tuning using other techniques of pseudo labels as alternative to PLL as already evaluated in ablation study at each 't'.

Also if it is assumed number of novel classes is known a priori, consider an ablation study to analyze sensitivity i.e. if our estimate of novel classes is off by some value at each time step 't' and in  addition these details could potentially be better explained by stating them explicitly in both setting and experiments.

Writing:
	In section 2.1 it is stated only unlabeled data is available and in line 60 it is stated small dataset is available for growing phase, please be explicit for readers to get a clear understand of setting. Is it source-free or exemplar only or some small replay buffer, etc.
	In line 43, 44 it is stated known classes classification and novelty detection would need different types of features, and this is not necessarily true or would suggest rewriting  sentence. As ideally we need more discriminative features to classify known + unknown and only training only w.r.t known may not have sufficient features to classify unknown but both need better discriminative features as indicated by [2] (closed set accuracy is all you need for open set)
	Minor corrections: Line 266, Phrase should be replaced with phase.

References:
1. Generalized Category Discovery https://openaccess.thecvf.com/content/CVPR2022/papers/Vaze_Generalized_Category_Discovery_CVPR_2022_paper.pdf
2. OPEN-SET RECOGNITION: A GOOD CLOSED-SET CLASSIFIER IS ALL YOU NEED? https://openreview.net/pdf?id=5hLP5JY9S2d

---

> ### Author Response · Authors · 2022-08-02
> **Response to Reviewer oK8E**
>
> We thank the reviewer for the careful reading and valuable feedback to our submission. We are encouraged that the reviewer found our proposed setting is important and practical and our framework generic and potentially be applied with various techniques. We sincerely apologize for the unclear presentation. Here we provide more experimental results and detailed description to demonstrate the effectiveness of GM. We hope our responce could solve your concerns.
>
> **[Concern 1]** About the GCD: We perform ablation study, which removes $\mathcal{L}_\text{BCE}$ +WTA and the cluster head, performs the supervised/unsupervised  contrastive loss in GCD on the exemplar set/unlabeled samples respectively, and applies semi-supervised K-Means algorithm for label assignment. The results in CI scenario and CIFAR-100 dataset are as follows::
>
> |                      | $\mathcal{M}_f$ | $\mathcal{M}_d$ |
> | -------------------- | --------------- | --------------- |
> | GM                   | 9.79            | 35.77           |
> | replace WTA with GCD | 7.43            | 30.03           |
>
> Compared to WTA, method in GCD obtain comparable performance (better $\mathcal{M}_f$ and worse $\mathcal{M}_d$), which are included in the revised version. More comparisons under DI, MI, SMI setting will be included in final version.
>
> **[Concern 2]** About number of novel classes: We have conducted experiments without knowing the number of new categories. Estimating the number of novel classes can be easily employed in GMNet. Please see feedback in the summary of the paper revision for more details.
>
> **[Concern 3]** About a small set of training examples (writing in line 60, 62): During the continuous category discovery stage, the newly added data is unlabeled, but a small dataset are maintained as the exemplar, which contains some labeled data from the initial stage and some pseudo-labeled data from previous CCD stage. This small part of data is also available in the CCD stage. We follow iCaRL and set the size of exemplar to 2000. It is noted that the prototypes, which represents the center of features of the exemplars in each class, are determined by the exemplar set. The exemplar set (and the prototypes) are used for
>
> During training:
>
> - [Growing phase] Novelty detection, in which the distance between the samples and the prototypes are calculated in order to detect novel samples.
> - [Merging phase] Pseudo label representation learning, in which the $\mathcal{L}_\text{PLL}$ is calculated based on the features of prototypes.
> - [Growing & Merging phase] Computation of $\mathcal{L}_\text{SD}$, which transfers the representation of both exemplars and current samples from the static branch to the dynamic branch.
>
> During inference:
>
> - [Merging phase] Label assigment, in which the (pseudo) labels are assigned according to the distance between the prototypes and the samples.
>
> Besides, to remove the restriction that the number of new classes needs to be known, during training and inference we also add estimating the number of novel classes in growing phase, where the semi-supervised k-means method is applied on both the exemplar set and the novel samples for estimition.
>
> **[Concern 4]** About the features needed by classification and novelty detection (writing in line 43, 44): Thanks for pointing this out. We modify the statement here to say "it turns out that these two task models usually produce different types of features: discriminative features on known classes are preferred by classification model}, while rich and diverse features are critical for identifying new classes"

---

> ### Author Response · Authors · 2022-08-06
> **A Gentle Reminder of Feedbacks**
>
> Dear Reviewer oK8E,
>
> Thanks again for your careful reading and valuable comments to improve our submission. We want to leave a gentle reminder due to the closing end time of the discussion period. We have tried to address all your concerns with detailed explanations and results, and revised the paper correspondingly. We would really appreciate feedback to make sure the responses and revisions have addressed all your concerns, or whether there is any leftover concern we can address.
>
> Sincerely
> Authors of Paper5057

---

### Author Response · Authors · 2022-08-02
**Summary of the Paper Revision (2/2)**

We also present a brief summary of our paper as follows:

- **Motivation**: We study a new problem of **Continuous Category Discovery (CCD)** requiring the model to discover novel categories in the data stream with satisfying performance on known classes, which is a practical problem in the application of machine learning systems and remains underexplored.

- **Method**: we develop a framework of **Grow and Merge (GM)** that works by alternating between a growing phase and a merge phase: in the growing phase, it increases the diversity of features through a continuous self-supervised learning for effective category mining, and in the merging phase, it merges the grown model with a static one to ensure satisfying performance for known classes.


- **Evaluation & contributions**: We investigate four experimental setting for the CCD problem and conduct extensive studies to verify that the proposed GM framework is significantly more effective than the state-of-the-art approaches for continuous category discovery on multiple datasets. We believe that the CCD problem is a practical problem worthy of study, and the proposed GM can be used as a basic framework for further research on the CCD problem.

---

### Author Response · Authors · 2022-08-02
**Summary of the Paper Revision (1/2)**

We thank all the reviewers for the careful reading and valuable feedback. We are encouraged that the reviewers thought our studied CCD problem to be of interest for the community (Reviewer kWya), important and practical (Reviewer oK8E), a natural and interesting extension of NCD (Reviewer WTKt), the proposed GM framework is useful and generic (Reviewer oK8E), and the experimental setting is properly defined (Reviewer kWya) with good analysis (Reviewer oK8E). We summarize the main concerns of the reviewers with the corresponding paper revision as follows.


**[Additional analysis and experiment results]**

1. About the number of novel classes during CCD

   Thanks to the reviewers for their suggestions. We propose Grow and Merge framework to solve the CCD problem. GM is a generic framework which focuses on how to implementing continuous classfication and novel classes discovery at the same time. Estimating the number of novel categories has been studied in literatures including [1][2], which can be easily employed in GMNet. The experimental results are provided in below:

   |                   | Class Estimated for $t=2$ | Class Estimated for $t=3$ | Class Estimated for $t=4$ | $\mathcal{M}_f$ | $\mathcal{M}_d$ |
   | ----------------- | ------------------------- | ------------------------- | ------------------------- | --------------- | --------------- |
   | CIFAR-100 (CI)    | 11                        | 13                        | 13                        | 10.29 (+0.5)    | 33.30 (-2.47)   |
   | ImageNet-100 (CI) | 12                        | 13                        | 13                        | 8.60 (+0.60)    | 25.47 (-1.33)   |
   | CIFAR-100 (MI)    | 14                        | 13                        | 13                        | 9.13 (-0.70)    | 27.70 (-4.71)   |
   | CIFAR-100 (SMI)   | 14                        | 14                        | 14                        | 9.21 (+0.37)    | 33.77 (-3.46)   |

   The values in the bracket represent the different of the performaces compared with the results with known number of categories. From the table and Table 2-4, we can find that GM without knowing the ground truth number of novelty classes can still outperform other methods using ground truth number of novelty classes. GM model only has 1.33 and 2.47 decreases under CI scenario, and no more than 4.71 decreases under more chanlleging MI and SMI scenarios, which is acceptable.

   [1] Learning to Discover Novel Visual Categories via Deep Transfer Clustering, ICCV 2019

   [2] Automatically Discovering and Learning New Visual Categories with Ranking Statistics, ICLR 2020

2. Additional Experimental results on more datasets and comparison method.

   1. Section 4.1: two datasets and one comparison method under CI scenario
   2. Section 4.2: one comparison under MI and SMI scenarios

3. Additional analysis of the GM model under different situation.

   1. Appendix E.1: estimating the number of novel categories
   2. Appendix E.3: analysis of the number of incremental classes at each time step

4. Adding experimental results with different data splits Appendix E.2.

5. Adding additional ablation studies in Appendix E.4.

6. Adding quantitative analysis of the feature distribution under CI on CIFAR-100 in Appendix E.5.

**[Revision of writings and descriptions]**

Based on the concerns of the reviewers, we revised the writings and descriptions in paper for better understanding, which are colored in blue in the revised version.

1. *Section 1 line 43*: clarifying the requirements of different tasks to feature distribution.

2. *Section 1 line 60*: modifying the brief summary of the proposed method to avoid confusions with the follow sections.

3. *Section 3*: re-organizing the method part for more clear presentation, including re-writing complicated descriptions, supplementing detailed definition of concepts to be self-contained, emphasizing the key components and hiddening some less import details.

4. *Figure 1, 3 & 4*: improving the captions to describe the illustration more clearly, such as the CCD setting and the meaning of dots in feature visualization.

---

### Author Response · Authors · 2022-08-09
**Summary of the Discussion**

Dear Chairs and Reviewers,

Hope this message finds you well.

With the closing of the discussion period, we present a brief summary of our discussion with the reviewers as an overview for reference.

First of all, we thank the reviewers for their careful reading and valuable feedback.  We are encouraged that the reviewers approved our rebuttal, and hope that our rebuttal could also address the remaining concerns of other reviewers.

---

Secondly, we present a brief summary of our contributions as follows:

- **[Problem]** We study a new problem of **Continuous Category Discovery (CCD)** requiring the model to discover novel categories in the data stream with satisfying performance on known classes, which is a practical problem in the application of machine learning systems and remains underexplored.
- **[Framework]** We develop a framework of **Grow and Merge (GM)** with key component Static-Dynamic Branches for effective category discovery for novel classes and performance maintenance for known classes. Extensive experiments demonstrate that the proposed framework outperforms recent methods and their combinations.


- **[Evaluation]** We investigate four experimental scenarios for the real-world applications with complicated data distribution for the CCD problem, *i.e.*, Class Incremental Scenario (CI), Data Incremental Scenario (DI), Mixed Incremental Scenario (MI), and Semi-supervised Mixed Incremental Scenario (SMI). We also design two core metrics to evaluate the capability of maintaining performances on known classes and discovering novel categories.
- **[Benchmark]** We conduct extensive studies to verify the effectiveness of the proposed GM framework for CCD on multiple datasets, including CIFAR-100, CUB-200, ImageNet-100, Stanford-Cars, FGVC-Aircraft, and ImageNet-200, under four scenarios, and show the superiority of the proposed framework compared with recent methods.

---

Thirdly, we summarize the main concerns of the reviewers with the corresponding paper revision as follows.

**[Additional analysis and experiment results]**

1. **Adding experiments about the number of novel classes during CCD** in the supplementary material F.1 and Table F1. We will move this part into the main paper if accepted because of the additional one page.
2. **Additional Experimental results on more datasets and comparison method** in Section 4.
   1. *Table 2*: add three datasets and one comparison method under CI scenario
   2. *Table 4*: add one comparison method under MI and SMI scenarios
3. **Additional analysis of the number of incremental classes at each time step** in the supplementary material F.2 and Table F2.
4. **Additional results with different data splits** in Tables 2-6, F1-F3, F7-F8.
5. **Adding additional ablation studies** in the supplementary material.
   1. Alternative of $\mathcal{L}_\text{BCE}$ with WTA hash in Table F3.
   2. Alternative of sample sifting strategy and ablation studies on the corresponding hyperparameter in Table F3 and Table F4.
   3. Ablation studies on the weights of each term in the loss function in Table F3 and Table F5.
   4. Ablation studies on the hyperparameter of novelty detection in Table F6.
   5. Ablation study and discussion about the effectiveness of GMNet in Table F7.
6. **Adding quantitative analysis of the feature distribution of different models** in Table F8.

**[Revision of writings and descriptions]**

Based on the concerns of the reviewers, we revised the writings and descriptions in the paper for better understanding.

1. *Section 1*: clarifying the description to avoid confusion.
2. *Section 3*: re-organizing the method part for more clear presentation.
3. *Figure 1, 3 & 4*: improving the captions to describe the illustration more clearly.
4. *Supplementary B & Figure B1*: adding a detailed pipeline of the GM framework with description.

---

Thanks again for your efforts in reviewing and discussing. We appreciate all the valuable suggestions and feedback that help us to improve our submission.

Yours sincerely,

Authors of Paper5057

---

### Meta-Review · Area_Chair_YoJg · 2022-08-24

**Recommendation:** Accept
**Confidence:** Certain

**Metareview:**

This paper proposes a method for continuous category discovery in novel category discovery tasks, assuming a dynamic setting in which unlabeled data are continuously fed to the model for category discovery. Specifically, it is a technique that balances the discovery of new categories with the merging of newly discovered information with knowledge already known to the model, and has shown its usefulness experimentally.  These results contribute to the progress of research in this field.

**Award:**

No

---

### Decision · Program_Chairs · 2022-09-14

Accept